# Single methyl groups can act as toggle switches to specify transmembrane Protein-protein interactions

Li He[1], Helena Steinocher[2], Ashish Shelar[1], Emily B Cohen[1†], Erin N Heim[1], Birthe B Kragelund[2], Gevorg Grigoryan[3], Daniel DiMaio[1,4,5,6]*

[1]Department of Genetics, Yale School of Medicine, New Haven, United States; [2]Department of Biology, Structural and NMR Laboratory, University of Copenhagen, Copenhagen, Denmark; [3]Department of Computer Science, Dartmouth College, Hanover, United States; [4]Department of Therapeutic Radiology, Yale School of Medicine, New Haven, United States; [5]Department of Molecular Biophysics & Biochemistry, Yale School of Medicine, New Haven, United States; [6]Yale Cancer Center, New Haven, United States

**Abstract** Transmembrane domains (TMDs) engage in protein-protein interactions that regulate many cellular processes, but the rules governing the specificity of these interactions are poorly understood. To discover these principles, we analyzed 26-residue model transmembrane proteins consisting exclusively of leucine and isoleucine (called LIL traptamers) that specifically activate the erythropoietin receptor (EPOR) in mouse cells to confer growth factor independence. We discovered that the placement of a single side chain methyl group at specific positions in a traptamer determined whether it associated productively with the TMD of the human EPOR, the mouse EPOR, or both receptors. Association of the traptamers with the EPOR induced EPOR oligomerization in an orientation that stimulated receptor activity. These results highlight the high intrinsic specificity of TMD interactions, demonstrate that a single methyl group can dictate specificity, and define the minimal chemical difference that can modulate the specificity of TMD interactions and the activity of transmembrane proteins.
DOI: https://doi.org/10.7554/eLife.27701.001

*For correspondence:
daniel.dimaio@yale.edu

Present address: †Beth Israel Deaconess Medical Center, Harvard Medical School, Boston, United States

Competing interests: The authors declare that no competing interests exist.

## Introduction

Specific interactions between proteins underlie much of biology. Proteomics analysis has identified thousands of protein-protein interactions (*e.g.*, [*Bork et al., 2004*]), but in most cases the molecular basis for the ability of a protein to bind to specific protein partners but not to closely related proteins is poorly understood. Integral membrane proteins constitute ~30% of the proteome of eukaryotic cells and serve many important biological functions (*Lehnert et al., 2004*). Transmembrane (TM) proteins contain TM domains (TMDs) that span the lipid bilayer. In multi-pass TM proteins, which span the membrane multiple times, TMDs interact to properly fold the protein into a functional state within the bilayer. For single-pass TM proteins, TMDs have traditionally been viewed as primarily serving to anchor the protein into the lipid bilayer, but they can engage in highly specific intermolecular protein-protein interactions that regulate protein oligomerization and activity (*Langosch and Arkin, 2009*; *Moore et al., 2008*). TMDs can align laterally in the membrane and oligomerize via hydrogen bonds, salt-bridges, or van der Waals packing interactions (*Langosch and Arkin, 2009*; *Moore et al., 2008*; *Bugge et al., 2016*; *Choma et al., 2000*; *Zhou et al., 2000*). However, the analysis of TMD complexes has been hindered by the difficulties in obtaining high-resolution structures of protein segments that cross membranes (*Bugge et al., 2016*).

Despite these limitations, TMDs also present advantages to the study of protein-protein interactions because they are short and usually adopt simple, helical structures in the membrane, which may fold independently of linked globular domains. The functional relevance of a free-standing TMD is illustrated by the E5 oncoprotein of bovine papillomavirus (BPV), a homodimeric, single-pass 44-residue TM protein that binds specifically to the TMD of the platelet-derived growth factor β receptor (PDGFβR), resulting in sustained receptor activation and cell transformation (*Lai et al., 1998*; *Petti et al., 1991*; *Schlegel et al., 1986*; *Goldstein et al., 1992*; *Petti and DiMaio, 1992*; *Talbert-Slagle and DiMaio, 2009*).

To study the interactions between TMDs, we developed a genetic system to isolate small (<50 residues), biologically active artificial TM proteins named traptamers (*Freeman-Cook and DiMaio, 2005*). In this approach, we construct libraries expressing many different traptamers with randomized hydrophobic segments, and use genetic selection to isolate rare clones with a desired biological activity in mammalian cells. We have isolated traptamers that activate the PDGFβR and cause morphologic transformation of fibroblasts; others that down-regulate expression of the HIV co-receptor, CCR5, and confer resistance to R5-tropic strains of HIV; and still others that activate the human erythropoietin receptor (hEPOR) and induce cell proliferation and erythroid differentiation of hematopoietic progenitor cells (*Cammett et al., 2010*; *Chacón et al., 2014*; *Freeman-Cook et al., 2004*, *2005*; *Scheideman et al., 2012*).

The EPOR is a single-pass TM protein that exists in a monomer-dimer equilibrium in unstimulated cells and is driven toward a productive dimeric form by EPO binding (*Constantinescu et al., 2001*; *Ebie and Fleming, 2007*; *Kubatzky et al., 2001*; *Livnah et al., 1999*; *Moraga et al., 2015*; *Ruan et al., 2004*). Ligand-induced dimerization of the EPOR in the proper orientation results in autophosphorylation of constitutively bound JAK2 kinases, phosphorylation of the intracellular domain of the EPOR, and initiation of intracellular signaling cascades (*Seubert et al., 2003*; *Syed et al., 1998*; *Witthuhn et al., 1993*; *Remy et al., 1999*; *Watowich, 2011*). The mouse EPOR (mEPOR) but not the hEPOR is also activated by gp55-P, a 55 kDa viral TM protein with a large, glycosylated extracellular domain (*Li et al., 1990*). Point mutations in the TMDs of gp55-P and the mEPOR can prevent EPOR activation (*Amanuma et al., 1989*; *Chung et al., 1989*; *Zon et al., 1992*; *Constantinescu et al., 1999*). However, no gp55-P mutants displayed altered specificity, so the features of gp55-P that allow it to distinguish between the closely related mEPOR and hEPOR are unknown, and the role of extra-membrane segments of gp55-P has not been explored.

We reasoned that by shortening and drastically reducing the chemical complexity of traptamers as model TM proteins, we could determine the minimal chemical complexity required to construct a biologically active protein. Remarkably, we discovered that 26-residue traptamers consisting of only leucines and isoleucines following an initiating methionine (LIL traptamers) can specifically activate the PDGFβR and transform cells (*Heim et al., 2015*). Because these proteins are comprised exclusively of two hydrophobic amino acids that differ chemically only by the placement of a single side chain methyl group and cannot undergo post-translational modification, these results defined the minimal chemical complexity required to generate an active protein.

Here, we report the isolation and characterization of similar LIL traptamers that activate the mEPOR, the hEPOR, or both receptors. The active traptamers associated with the TMD of the target EPOR and induced EPOR oligomerization in an orientation that supported signaling. Most strikingly, the placement of a single methyl group at precise positions in the traptamer dictated its differential ability to associate with and activate the hEPOR or the mEPOR. These results reveal principles that govern the formation of an important class of protein-protein interactions by defining the minimal chemical difference that can determine specificity.

## Results

### Isolation of LIL traptamers that cooperate with the erythropoietin receptor

To isolate LIL traptamers that activate the EPOR, we screened the UDv6 traptamer library for proteins that support interleukin-3 (IL-3)-independent proliferation of BaF3 cells engineered to express the HA-tagged hEPOR (BaF3/hEPOR cells) (*Figure 1A*). Murine BaF3 cells are dependent on IL-3 for survival and proliferation, and traptamers that activate the EPOR allow BaF3/hEPOR cells to

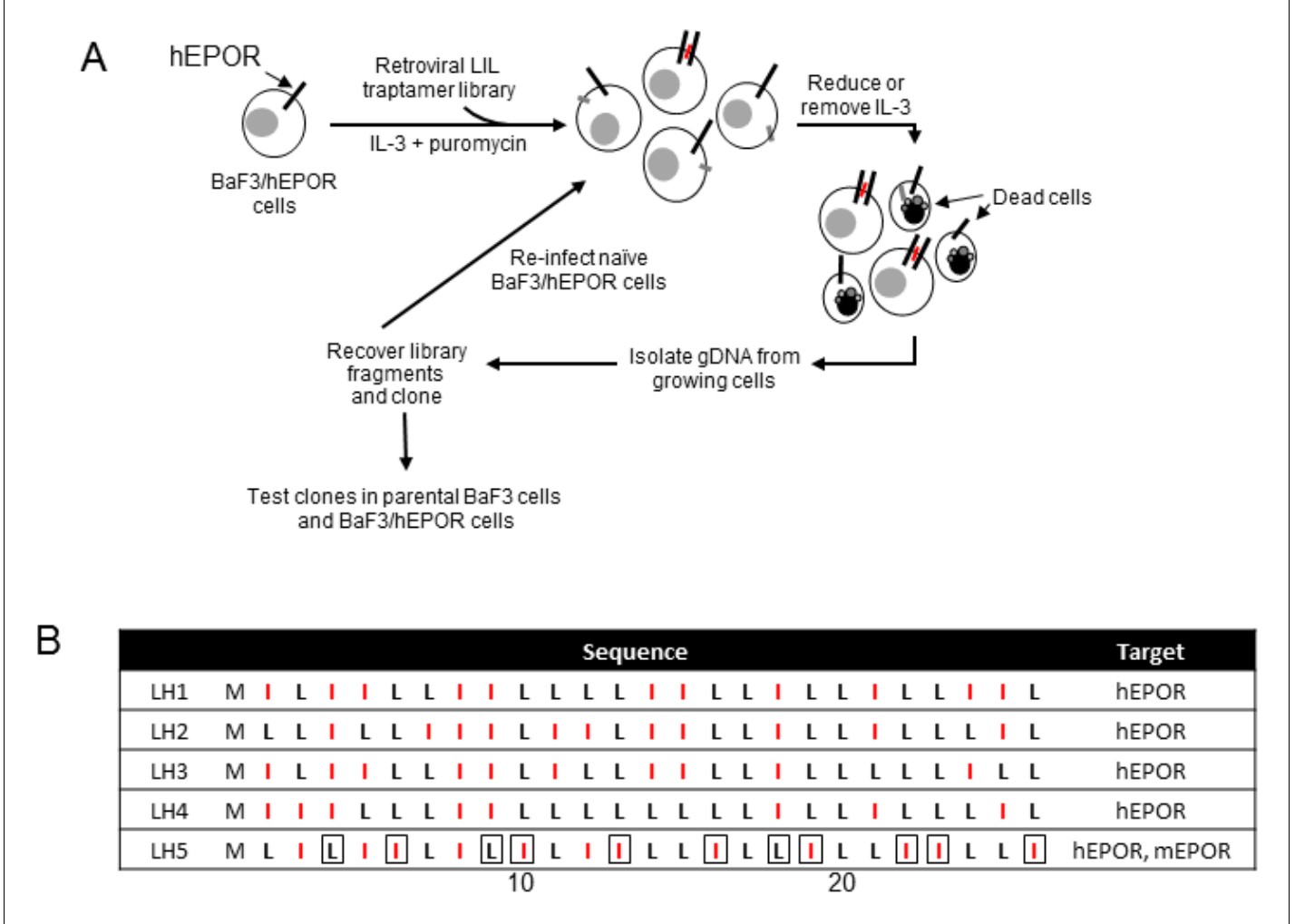

**Figure 1.** Isolation of traptamers that activate EPOR. (**A**) Scheme to isolate traptamers that activate the hEPOR. BaF3/hEPOR cells were infected with a retrovirus library expressing LIL traptamers and selected in medium containing a reduced level of IL-3. Genes encoding traptamers were recovered from genomic DNA isolated from live cells, cloned, re-introduced into naïve BaF3/hEPOR cells and subjected to an additional round of selection in the absence of IL-3. (Red bars, traptamers that activate hEPOR; grey bars, traptamers that do not activate hEPOR.) (**B**) Sequences of traptamers that confer IL-3 independence in BaF3/hEPOR cells but not in parental BaF3 cells. M, methionine; L, leucine; I, isoleucine (colored red). Residues in boxes are distinctive for LH5. Numbers at bottom indicate the position in the traptamer sequence.

DOI: https://doi.org/10.7554/eLife.27701.002

proliferate in the absence of IL-3 (*Cammett et al., 2010*; *Li et al., 1990*). The UDv6 library expresses LIL traptamers in which a methionine initiation codon is followed by 25 codons encoding equimolar leucine and isoleucine in a random order (*Heim et al., 2015*). The vast majority of LIL traptamers in UDv6 did not confer IL-3 independence.

Individual LIL traptamers were cloned from IL-3 independent cells and tested for their ability to confer IL-3 independence in BaF3 cells lacking hEPOR expression, in BaF3/hEPOR cells, and in BaF3 cells expressing HA-tagged mouse EPOR (BaF3/mEPOR). Cells were infected with retroviruses expressing a traptamer or with control MSCVp retrovirus vector, selected with puromycin, and incubated in IL-3-free media. Five LIL traptamers stimulated IL-3-independent growth of BaF3/hEPOR cells but not parental BaF3 cells (*Figure 2A*), demonstrating that the EPOR was required for the activity of these traptamers. Although traptamers LH1, LH2, LH3, and LH4 cooperated with the hEPOR, they did not cooperate with the mEPOR, highlighting the high specificity of these traptamers. In contrast, LH5 cooperated with both the mEPOR and the hEPOR. The sequences of these traptamers (shown in *Figure 1B*) are unrelated to the more complex sequences of the 44-residue

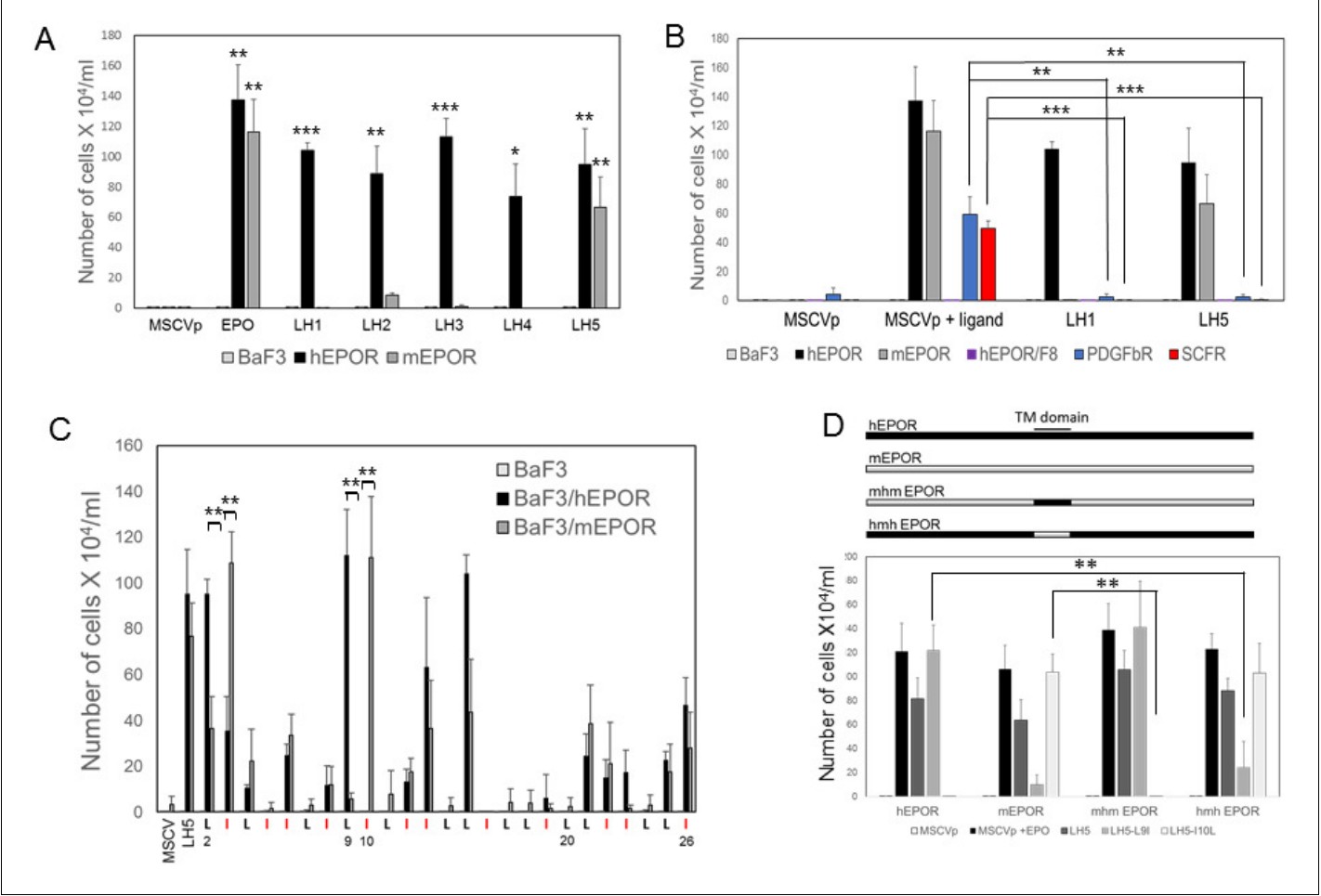

**Figure 2.** Specificity of traptamer action. (**A**) BaF3, BaF3/hEPOR and BaF3/mEPOR cells stably expressing empty MSCVp vector or the indicated traptamer were incubated in medium lacking IL-3. MSCVp cells were also incubated in the presence of EPO, as indicated. The number of live cells was counted six days after IL-3 removal. The bars show mean ± SEM for three independent experiments. For clarity, in all graphs, values for samples with no growth were arbitrarily set at 0.5 × 10⁴ cells/ml. Statistical significance of differences between cells expressing MSCVp and cells expressing a traptamer were evaluated by two-tailed Student's t-test with unequal variance (*p≤0.05; **p≤0.02; ***p≤0.005). (**B**) BaF3 cells and BaF3 cells expressing wild-type hEPOR, F8 (a phosphorylation-defective mutant of the hEPOR), mEPOR, PDGFβR or SCFR were infected with MSCVp or MSCVp expressing LH1 or LH5. After puromycin selection, cells were incubated in medium lacking IL-3. Cells expressing MSCVp were also treated with the cognate ligand, as indicated: EPO for EPOR and F8, PDGF for PDGFβR, and stem cell factor for SCFR. The number of live cells was counted six days after IL-3 removal. The bars shown mean ± SEM for three independent experiments. Statistical significance of differences between cells treated with ligand and cells expressing a traptamer were evaluated by two-tailed Student's t-test with unequal variance (**p≤0.02; ***p≤0.005). (**C**) Sequence of LH5 starting at position two is shown at the bottom, with isoleucines colored red. Each leucine was mutated individually to isoleucine and each isoleucine was mutated to leucine. BaF3, BaF3/hEPOR and BaF3/mEPOR cells were infected with MSCVp or MSCVp expressing LH5 or one of the LH5 mutants. After puromycin selection, cells were incubated in medium lacking IL-3. In cells expressing a traptamer with a mutation at the indicated position, the number of live cells was counted six days after IL-3 removal. The bars show mean ± SEM for three independent experiments. For each mutant, statistical significance of differences between cells expressing hEPOR and cells expressing mEPOR were evaluated by two-tailed Student's t-test with unequal variance (**p≤0.02). (**D**) **Top panel**. Schematic diagram of mhm and hmh chimeric receptors with hEPOR and mEPOR segments shown in black and gray, respectively. **Bottom panel**. BaF3 cells expressing the hEPOR, the mEPOR, or a chimeric EPOR were infected with MSCVp or MSCVp expressing the indicated traptamer. After selection with puromycin, cells were tested for IL-3 independence. The number of live cells was counted six days after IL-3 removal. The bars show mean ± SEM for three independent experiments. Statistical significance of differences between cells expressing wild-type hEPOR or mEPOR compared to cells expressing a chimeric EPOR were evaluated by two-tailed Student's t-test with unequal variance (**p≤0.02).

DOI: https://doi.org/10.7554/eLife.27701.003

traptamers reported earlier that specifically activate the hEPOR (*Cammett et al., 2010*; *Cohen et al., 2014*). LI-5 (a traptamer that cooperates with the PDGFβR) and several unselected LIL traptamers did not confer IL-3 independence in BaF3/hEPOR cells (data not shown).

Neither LH1 nor LH5 induced IL-3 independence in cells expressing hEPOR/F8, a phosphorylation-defective mutant of the hEPOR that does not support EPO-induced proliferation (*Wu et al., 1997*) (*Figure 2B*), suggesting that activation of the EPOR and its downstream signaling pathways were required for traptamer-induced cell proliferation. To assess specificity, we also co-expressed LH1 and LH5 with the PDGFβR or the stem cell factor receptor (SCFR), two receptor tyrosine kinases unrelated to the EPOR. Neither traptamer induced IL-3 independence in cells expressing these receptors, even though these cells proliferated in response to their cognate ligands (*Figure 2B*). Thus, the LIL traptamers studied here specifically activated the EPOR.

## Mutational analysis of LH5 identified residues important for activity and specificity

To explore the basis for traptamer specificity, we conducted a comprehensive mutational analysis of LH5 by individually replacing each isoleucine with leucine and each leucine with isoleucine, and tested the activity of the resulting 25 point mutants in parental BaF3 cells or BaF3 cells expressing the hEPOR or the mEPOR (*Figure 2C*). None of the mutants were active in parental BaF3 cells. Several mutants, for example LH5-I5L, LH5-L14I, and LH5-I16L, failed to cooperate with either EPOR, demonstrating that a leucine or an isoleucine at particular positions is required for activity. Most interestingly, the substitutions L9I and I10L dramatically changed the specificity of LH5: LH5-L9I cooperated with the hEPOR and not the mEPOR, and LH5-I10L cooperated with the mEPOR but not the human receptor. These results demonstrate that highly specific protein-protein interactions can be fine-tuned by changing the position of a single side chain methyl group at specific positions in a short TM protein with minimal chemical complexity.

## Traptamer specificity is determined by the EPOR TMD

Given the entirely hydrophobic nature of LIL traptamers, we hypothesized that their specificity was due to their ability to recognize sequence differences in the TMD of the EPORs. To assess if the TMD of the EPOR determined whether it responded to the LH5 mutants, we constructed two chimeric EPORs by swapping the TMDs of hEPOR and mEPOR and tested the ability of the chimeras to respond to various traptamers. In the mhm chimera, the TMD of the mEPOR is replaced with the TMD of hEPOR, and in the hmh chimera, the TMD of the hEPOR was replaced with the TMD of mEPOR (*Figure 2D*, top). EPO and LH5 induced IL-3 independence in BaF3 cells expressing either of these chimeras (*Figure 2D*, bottom panel), consistent with the ability of EPO and LH5 to activate both hEPOR and mEPOR. LH5-I10L, which is specific for the mEPOR, cooperated with hmhEPOR but not with mhmEPOR, whereas LH5-l9L, which is specific for hEPOR, cooperated with mhmEPOR and poorly with hmhEPOR. Thus, the ability of the traptamers to discriminate between the hEPOR and the mEPOR was determined by the amino acid sequence of the EPOR TMD.

## Activation of EPOR and JAK2 by LIL traptamers

Tyrosine phosphorylation of the EPOR and its associated JAK2 kinases is a hallmark of EPOR activation. To determine if the traptamers induced tyrosine phosphorylation of EPOR, lysates were prepared from BaF3/hEPOR and BaF3/mEPOR cells expressing an empty vector, LH5, or one of its mutants. Lysates were subjected to anti-HA immunoprecipitation to recover the HA-tagged EPOR, and immunoblotted with a broadly reactive anti-phosphotyrosine antibody. As shown in *Figure 3A*, there was little, if any, tyrosine phosphorylation of either hEPOR or mEPOR in cells expressing the empty vector (lane 1). EPO treatment (lane 2) or expression of LH5 (lane 3) caused tyrosine phosphorylation of both hEPOR and mEPOR. However, the two EPORs responded differently to the mutant traptamers. LH5-L9I caused tyrosine phosphorylation of hEPOR but not mEPOR (lane 4), and LH5-I10L (lane 5) caused tyrosine phosphorylation of mEPOR but not hEPOR. Thus, EPOR tyrosine phosphorylation correlated perfectly with the ability of the traptamers to induce growth factor independence.

Samples were also assayed by immunoblotting with an antibody that detects phosphorylation of JAK2 at Tyr1008 (*Figure 3B*). JAK2 phosphorylation displayed the same pattern as noted above for

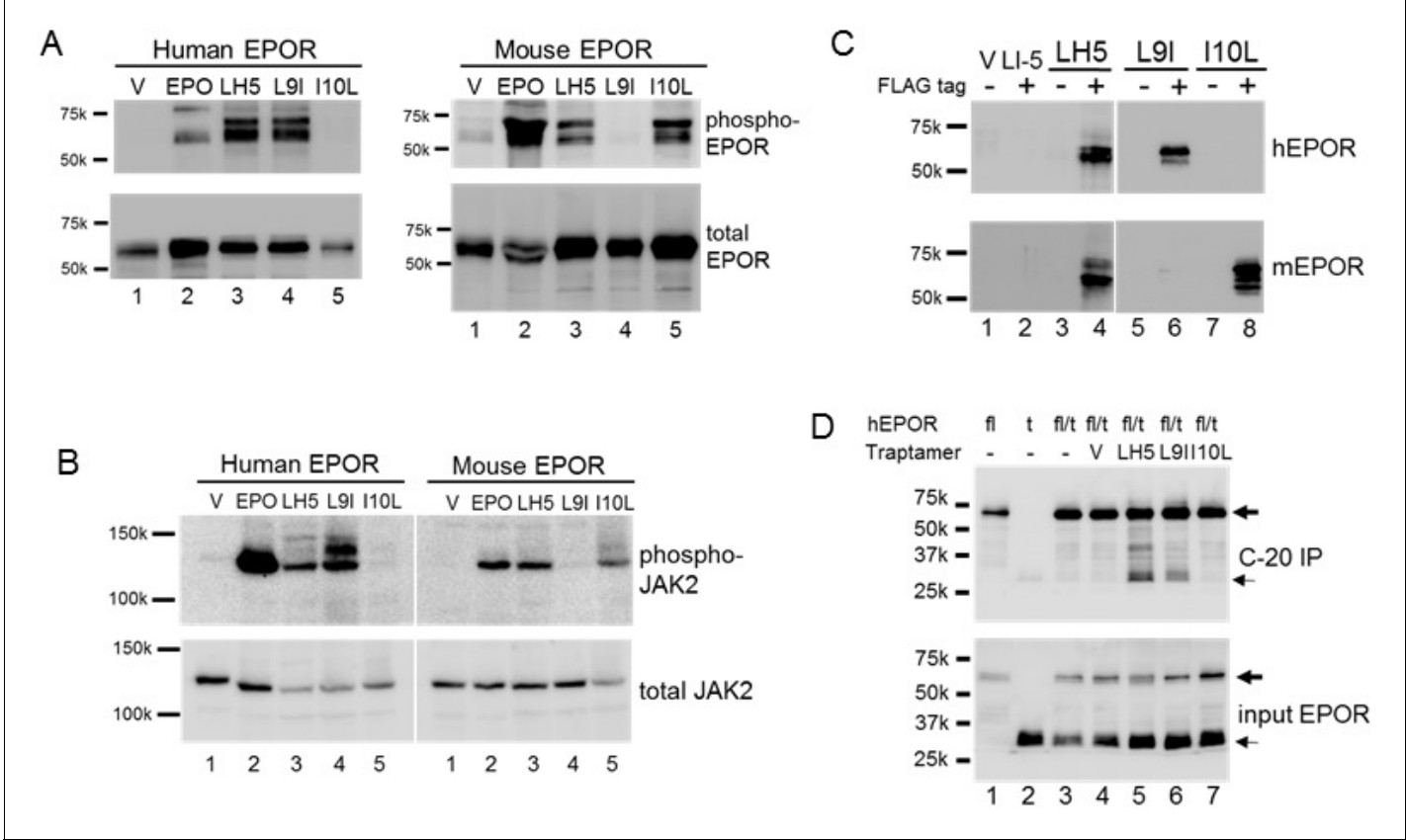

**Figure 3.** Biochemical analysis of EPOR activation. (**A**) Extracts were prepared from BaF3/hEPOR cells (left panels) or BaF3/mEPOR cells (right panels) expressing MSCVp (V), LH5, LH5-L9I (L9I), or LH5-I10L (I10L). MSCVp-expressing cells were also acutely treated with EPO, as indicated. Extracts were immunoprecipitated with anti-HA antibody and immunoblotted with anti-phosphotyrosine antibody PY100 (phospho-EPOR). The multiple species represent minor phosphorylated forms of the EPOR. The same blot was stripped and reprobed for total EPOR by using anti-HA antibody. (**B**) Cell extracts described in panel A were immunoblotted with anti-phospho-JAK2 antibody (phospho-JAK2). The same blot was stripped and reprobed for total JAK2 using anti-JAK2 antibody. Size of protein markers (in kDa) is shown in A and B. (**C**) Extracts were prepared from BaF3/hEPOR cells (top panel) or BaF3/mEPOR cells (bottom panel) expressing MSCVp vector (V), FLAG-tagged LI-5 (a traptamer that specifically activates PDGFβR [*Heim et al., 2015*]), or FLAG-tagged (+) or untagged (-) LH5, LH5-L9I, or LH5-I10L. Extracts were immunoprecipitated with anti-FLAG antibody and immunoblotted with anti-HA antibody. (**D**) Extracts of BaF3 cells expressing the full-length hEPOR (fl) and/or the Δ259 truncated hEPOR (t) and the indicated traptamer or MSCVp (V) were immunoprecipitated with anti-C-20 antibody, which recognizes only full-length EPOR, and immunoblotted with anti-HA antibody (C-20 IP) (top panel); or were directly immunoblotted with anti-HA antibody (input EPOR) (bottom panel). – indicates no traptamer. The full-length and truncated EPORs are indicated by the bold and thin arrows, respectively.

DOI: https://doi.org/10.7554/eLife.27701.004

The following figure supplements are available for figure 3:

**Figure supplement 1.** Expression of FLAG-tagged traptamers and HA-tagged EPOR.
DOI: https://doi.org/10.7554/eLife.27701.005

**Figure supplement 2.** Addition of a FLAG-tag does not change the specificity of traptamers.
DOI: https://doi.org/10.7554/eLife.27701.006

**Figure supplement 3.** Defective LH5 mutants do not bind hEPOR.
DOI: https://doi.org/10.7554/eLife.27701.007

**Figure supplement 4.** Active traptamers bind truncated hEPOR.
DOI: https://doi.org/10.7554/eLife.27701.008

EPOR. Namely, compared to unstimulated cells (lane 1), EPO treatment (lane 2) or expression of LH5 (lane 3) caused phosphorylation of JAK2 in cells expressing either hEPOR or mEPOR. In contrast, LH5-L9I induced JAK2 phosphorylation only in cells expressing the hEPOR (lane 4), and LH5-I10L induced phosphorylation of JAK2 only in cells expressing the mEPOR (lane 5). Taken together,

these data show that the traptamers specifically activate the EPOR signaling pathways to induce growth factor independence.

## Stable association between the active traptamers and the EPO receptor

We used co-immunoprecipitation to determine whether the traptamers formed a complex with the EPOR. We first added a FLAG epitope tag to the N-terminus of LH5, LH5-L9I, and LH5-I10L. FLAG-tagged traptamers were expressed in BaF3 cells together with hEPOR or mEPOR (*Figure 3—figure supplement 1*) and displayed the same specificity as their untagged counterparts (*Figure 3—figure supplement 2*). The activity of the tagged traptamers implies that they adopt a type II TM orientation (*i.e.,* N-terminus in the cytoplasm), because it is unlikely that the highly charged N-terminal FLAG tag can cross the membrane when the traptamers are expressed in cells.

To assess complex formation, detergent lysates of cells expressing various combinations of traptamers and EPORs were immunoprecipitated with anti-FLAG beads and then probed with anti-HA antibody to detect HA-tagged EPOR associated with FLAG-tagged traptamer (*Figure 3C*). No EPOR was immunoprecipitated from cells not expressing a traptamer (lane 1) or expressing an untagged traptamer (lanes 3, 5, and 7), demonstrating that the anti-FLAG antibody does not cross-react with the EPOR. Similarly, EPOR was not co-immunoprecipitated from cells expressing FLAG-tagged LI-5, which activates PDGFβR but not the EPOR (*Heim et al., 2015*) (lane 2), or from cells expressing FLAG-tagged defective mutants LH5-L14I or LH5-L20I (*Figure 3—figure supplement 3*), highlighting that these traptamers do not associate with the EPOR with sufficient affinity to be detected in this assay. In contrast, anti-FLAG antibody co-immunoprecipitated the HA-tagged hEPOR and mEPOR when FLAG-tagged LH5 was expressed in the cells (lane 4), showing that LH5 was indeed in a physical complex with the EPOR. Strikingly, the ability of the LH5 mutants to bind hEPOR or mEPOR in this assay correlated with their biological activity (*Figure 3C*): LH5-L9I bound to only hEPOR (lane 6), and LH5-I10L bound to only mEPOR (lane 8). These results strongly suggest that LIL traptamers induce EPOR activation through binding to the EPOR, and that the ability of the LH5 mutants to activate only one form of the EPOR is due to specific binding of the mutants to the hEPOR or mEPOR.

## Traptamers enhance oligomerization of EPO receptor

To determine whether the traptamers induced EPOR dimerization, we conducted co-immunoprecipitation experiments with BaF3 cells co-expressing full-length hEPOR and Δ259, a C-terminally truncated HA-tagged hEPOR mutant consisting of the extracellular ligand-binding domain and TMD, but lacking the entire cytoplasmic domain. LH5 and LH5-L9I retained the ability to bind Δ259, whereas LH5-I10L was defective, consistent with its inability to bind the hEPOR (*Figure 3—figure supplement 4*).

BaF3 cells expressing the full-length hEPOR, Δ259, or both full-length and truncated hEPOR were maintained in medium containing IL-3. For these co-immunoprecipitation experiments, we used C-20, an anti-hEPOR antibody that does not recognize Δ259 because of the deletion. As described in the methods, we confirmed that the C-20 antibody did not directly immunoprecipitate Δ259 and that under these conditions there is little basal dimerization between full-length and truncated EPOR. BaF3/Δ259-hEPOR cells were then infected with the empty MSCVp vector, or MSCVp expressing LH5, LH5-L9I or LH5-I10L. If a traptamer promotes hEPOR dimerization, Δ259 should be immunoprecipitated with anti-C-20 due to its association with full-length hEPOR. As shown in lanes 5 and 6 of *Figure 3D*, top panel, Δ259 was indeed observed in C-20 immunoprecipitates from cells expressing traptamer LH5 and LH5-L9I, the two traptamers that interact with hEPOR. In contrast, little if any Δ259 was co-immunoprecipitated from cells infected with MSCVp empty vector (lane 4) or expressing LH5-I10L (lane 7), which does not interact with the hEPOR. These results demonstrate that the two traptamers that activate the hEPOR, LH5 and LH5-L9I, induce hEPOR oligomerization, presumably dimerization, whereas LH5-I10L does not.

## NMR evidence that the active traptamers induce dimerization of the hEPOR TMD in detergent micelles

To test whether the traptamers affected oligomerization of the EPOR TMD, we used NMR to determine the effect of adding traptamers to the hEPOR TMD in 1,2-dihexanoyl-*sn*-glycero-3-phosphocholine (DHPC) detergent micelles. We used a 61-residue segment of the hEPOR (residues 217 to 277), comprising ~10 residues upstream of the TMD, the TMD itself, and ~30 residues downstream of the TMD. This segment (hEPOR$_{217-277}$) was labeled with $^{15}$N (or with $^{13}$C and $^{15}$N) and purified from bacteria (*Bugge et al., 2015*). A peptide containing the TMD of the unrelated human growth hormone receptor (hGHR) (residues 238 to 274) was also labeled and purified. Similarly, we purified recombinant unlabeled LH5, LH5-L9I, and LH5-I10L.

hEPOR$_{217-277}$ was reconstituted in DHPC micelles and its backbone NMR resonances were assigned (*Figure 4A*). Earlier work showed that the EPOR TMD weakly dimerizes in C14SB detergent micelles, and similar to other TMDs its monomer-dimer equilibrium is determined by the protein-to-detergent ratio (*Ebie and Fleming, 2007*; *Mineev et al., 2014*). Therefore, the hEPOR$_{217-277}$ monomer-dimer equilibrium was assayed by monitoring changes in protein backbone chemical shifts upon DHPC titration. The overlay of spectra at high (red) and low (black) DHPC concentrations showed clear changes in amide backbone chemical shifts (representative residues shown in *Figure 4B*). Thus, a change in equilibrium occurred, likely reflecting a switch of the hEPOR TMD to a more monomeric state at increasing DHPC. The weighted and combined nitrogen and proton chemical shift changes per residue (*Figure 4C*, red bars) revealed numerous residue-specific changes, most of which likely reflected packing differences with the detergent upon monomerization, most dramatically for residues close to the detergent head groups. Importantly, the chemical shift changes were not dependent on the concentration of DHPC in the sample but only on the ratio between protein and detergent (data not shown). Thus, we conclude that these residue-specific changes reflect the oligomerization state of the EPOR TMD.

To determine the effect of LH5 on the hEPOR TMD, the LH5 peptide was added at 1:2 and 2:2 molar ratios to hEPOR$_{217-277}$, while keeping the ratio of total protein to detergent constant. Addition of LH5 at low DHPC caused the chemical shifts of hEPOR$_{217-277}$ to change as illustrated by the NMR signals from Leu230 and Ala245 in the $^1$H,$^{15}$N-HSQC spectra (*Figure 4B*, blue peaks). The same resonances were affected by DHPC increase (red) or LH5 addition (blue), but in opposite directions (*Figure 4B and C*). Thus, these treatments most likely induce opposite changes in hEPOR$_{217-277}$, with DHPC pushing the equilibrium toward the monomer and LH5 pushing it toward the dimer. We did not observe specific LH5-induced chemical shift changes in the middle of the TMD where LH5 Leu9 and Ile10 appear to interact with the EPOR. However, the interactions of LH5 with hEPOR$_{217-277}$ might be primarily side chain directed, as suggested by the mutation data, and thus not observable by backbone amide chemical shifts. In this case, dimerization-induced detergent repacking would dominate the changes in chemical shift. To further test the effect of LH5 on hEPOR$_{217-277}$ side chain chemical shifts, we recorded $^{13}$C-HSQCs of $^{13}$C,$^{15}$N-hEPOR$_{217-277}$ alone and in complex with LH5. Overall the two spectra aligned very well and no major chemical shift changes were observed (*Figure 4—figure supplement 1A*). However, as shown in *Figure 4—figure supplement 1B*, addition of the traptamer caused the appearance of two novel peaks. Although we have not assigned these peaks, they do not appear to belong to hydrophobic side chain atoms, or a serine or threonine within the TMD, as the chemical shifts specific to the side chains of these residues were identical in the presence and absence of LH5. Regardless, these data show that the traptamers do change the chemical environment of a restricted number of side chains in the segment of the hEPOR included in these experiments.

When the mutant traptamer peptides were added to $^{15}$N-labeled hEPOR TMD, the same residues were affected in the same direction as when LH5 was added, but the amplitudes of the chemical shift changes in the $^{15}$N,$^1$H-HSQC spectra were different (*Figure 4—figure supplement 2*). In accordance with the inability of LH5-I10L to cooperate with the hEPOR, this peptide induced the smallest chemical shift changes, whereas LH5-L9I and LH5, which cooperate with hEPOR, induced larger changes, illustrated for Leu230 and Ala245 in *Figure 4D*. These experiments indicate that all three peptides enhance dimerization of the hEPOR TMD, but the LH5 and LH5-L9I peptides were more active at inducing dimerization, consistent with the ability of the corresponding traptamers to cooperate with the hEPOR.

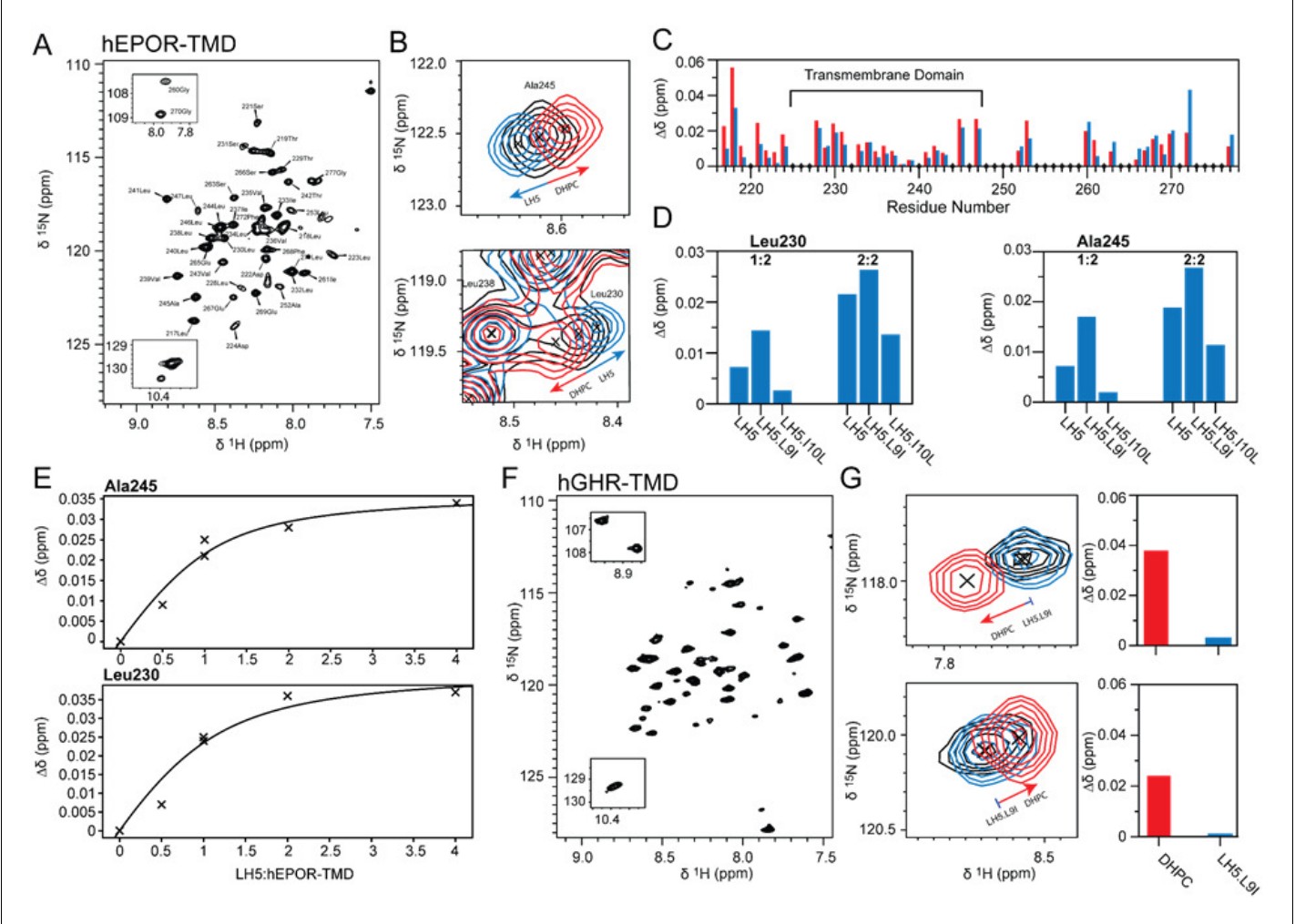

**Figure 4.** LH5 drives EPOR TMD peptides into dimers in detergent micelles. (**A**) $^1$H,$^{15}$N- HSQC spectrum of $^{15}$N-labelled hEPOR$_{217-277}$ in DHPC micelles with assignments indicated. (**B**) Zoomed-in contour plots of hEPOR Ala245 and Leu230 from $^1$H,$^{15}$N- HSQC spectra. Peaks from hEPOR$_{217-277}$ at 30-times molar excess of DHPC are depicted in black, peaks at 60-times molar excess of DHPC in red, and peaks with LH5 in a ratio of LH5:hEPOR:DHPC of 1:1:30 in blue. Increasing the amounts of DHPC or adding LH5 induced opposite effects as indicated by arrows. (**C**) Weighted chemical shift changes of $^{15}$N-hEPOR$_{217-277}$ backbone amides per residue upon increasing the molar excess of DHPC from 30 to 60 times (red bars) or addition of LH5 in a molar ratio of 1:1 (blue bars). The same residues were affected by LH5 and DHPC, but the peaks move in opposite directions. The hEPOR TMD is indicated by a black horizontal line. Diamonds on the x-axis indicate proline residues, overlapping peaks, and missing assignments. (**D**) Weighted chemical shift changes of Leu230 and Ala245 backbone amides induced by addition of LH5, LH5-L9I or LH5-I10L at molar ratios of one traptamer molecule to two hEPOR TMDs (1:2) and two traptamer molecules to two hEPOR TMDs (2:2). (**E**) Weighted chemical shift changes of $^{15}$N-hEPOR$_{217-277}$ Ala245 and Leu230 backbone amides induced by addition of LH5 at different molar ratios of LH5 to hEPOR$_{217-277}$ in DHPC micelles. Two individual titrations were performed, one from 0:1 to 1:1 (corresponding to the 2:2 traptamer:EPOR TMD ratio) and the other from 0:1 to 4:1 (corresponding to the 8:2 traptamer:EPOR TMD ratio) molar ratio of LH5 to hEPOR$_{217-277}$. The titration point at 1:1 was recorded in each series. A hyperbola is indicated to guide the eye. (**F**) $^1$H,$^{15}$N- HSQC spectrum of $^{15}$N-labelled hGHR$_{238-274}$ in DHPC micelles. (**G**) Zoomed-in contour plots of two selected peaks of hGHR from $^1$H,$^{15}$N- HSQC spectra. Peaks from hGHR$_{238-274}$ at 60-times molar excess of DHPC are depicted in black, peaks at 120-times molar excess of DHPC in red, and peaks with LH5-L9I in a ratio of LH5-L9I:hEPOR:DHPC of 2:1:60 in blue. Addition of LH5-L9I did not induce a change in chemical shifts as indicated by the near-complete overlap of the black and blue peaks (**Figure 4—figure supplement 3**). The extent of the chemical shift changes in the hGHR$_{238-274}$ spectra is illustrated in the bar graphs showing the changes induced by the DHPC in red and by LH5-L9I in blue for the peaks shown on the left.

DOI: https://doi.org/10.7554/eLife.27701.009

The following figure supplements are available for figure 4:

**Figure supplement 1.** hEPOR TMD side chain chemical shift.

DOI: https://doi.org/10.7554/eLife.27701.010

**Figure supplement 2.** LH5 mutants drive hEPOR TMD peptides into dimers.

*Figure 4 continued on next page*

*Figure 4 continued*

DOI: https://doi.org/10.7554/eLife.27701.011

**Figure supplement 3.** LH5-9LI does not change the growth hormone receptor monomer-dimer equilibrium.

DOI: https://doi.org/10.7554/eLife.27701.012

The differential ability of LH5-L9I and LH5-I10L to drive dimerization of the hEPOR TMD implies that this is a sequence-specific effect mediated by direct interactions between the TMDs, rather than by the traptamers changing the lipid/detergent environment of hEPOR TMD. If it is a direct effect, we predict that the traptamers would exert a saturable effect on the NMR spectra of the EPOR TMD, and additionally would not affect the TMD of an unrelated receptor. To test these predictions, we added traptamers at various ratios to $hEPOR_{217-277}$ and measured the induced chemical shift. As shown in Fig, 4E, the NMR shifts were clearly saturable at a traptamer-to-EPOR TMD ratio approaching 2 (as it would be for the 4:2 complex). Because of the small, induced chemical shift changes and the possibility of the experiment not being conducted at concentration well above $K_D$, we cannot assign a precise stoichiometry to the complex. Nevertheless, these results show that more than one and possibly four traptamer helices associate with each EPOR TMD dimer. Importantly, the saturability of this process shows that we observe a direct interaction rather than an indirect effect upon addition of LH5. We also investigated whether the equilibrium of the TMD of an unrelated receptor, the human growth hormone receptor (hGHR), was affected by LH5-L9I, the traptamer mutant that had the greatest effect on the hEPOR TMD. NMR spectra were obtained for the TMD of the hGHR in the presence and absence of LH5-L9I and at different protein-to-DHPC ratios (*Figure 4F and G*). As shown in *Figure 4G* and *Figure 4—figure supplement 3A*, DHPC titration had a marked effect on several hGHR chemical shifts. In contrast, LH5-L9I had little or no effect on the hGHR TMD chemical shifts (*Figure 4G* and *Figure 4—figure supplement 3B*), demonstrating that LH5 traptamers specifically stabilize hEPOR dimerization.

We also considered the possibility that the observed dimerization effect could be caused by LH5 titrating DHPC away from $hEPOR_{217-277}$, thereby indirectly shifting the monomer:dimer equilibrium towards the dimer. However, the leucine or isoleucine substitution in the traptamer mutants would not modulate the hydrophobicity of LH5 to an extent that this effect could account for the diverse chemical shift changes observed. In addition, it seems likely that such an indirect effect would also affect the hGHR TMD. We thus conclude that the traptamers directly stabilize the dimeric form of the hEPOR TMD.

## Residue at position 10 of traptamer LH5 plays an important role in determining specificity

We next sought to identify specific amino acids responsible for productive interaction between LH5 and the EPOR TMDs. We constructed LH5 mutants containing all combinations of leucine, isoleucine, valine and alanine at position 9 and 10 and tested the ability of these 16 proteins to cooperate with hEPOR or mEPOR in BaF3 cells. Valine is structurally similar to isoleucine except it lacks a terminal methyl group, and alanine has a small aliphatic side chain. The ability of a traptamer to cooperate with the hEPOR was determined solely by the presence of an isoleucine at position 10. Traptamers with an isoleucine at this position cooperated with the hEPOR, whereas mutants containing a leucine, valine, or alanine at position 10 displayed little if any ability to cooperate with the hEPOR (*Figure 5*, top panel). In contrast, the chemistry of the amino acid at position nine did not affect activity. Position 10 was also a prime determinant of activity with the mEPOR (*Figure 5*, bottom panel). Traptamers with leucine in position 10 cooperated with the mEPOR, regardless of the amino acid at position 9. Traptamers with isoleucine at position 10 activated the mEPOR only if they also contained a leucine or alanine at position 9. Traptamers with valine or alanine at position 10 did not cooperate with the mEPOR. Thus, the amino acid at position 10 was the prime determinant of specificity.

## A small amino acid at position 238 in the TMD of the mEPOR determines specificity

The sequence of the TMD of mEPOR differs from that of hEPOR at three positions, Leu236 in place of valine, Ser238 in place of leucine, and Leu239 in place of valine (the numbering throughout is

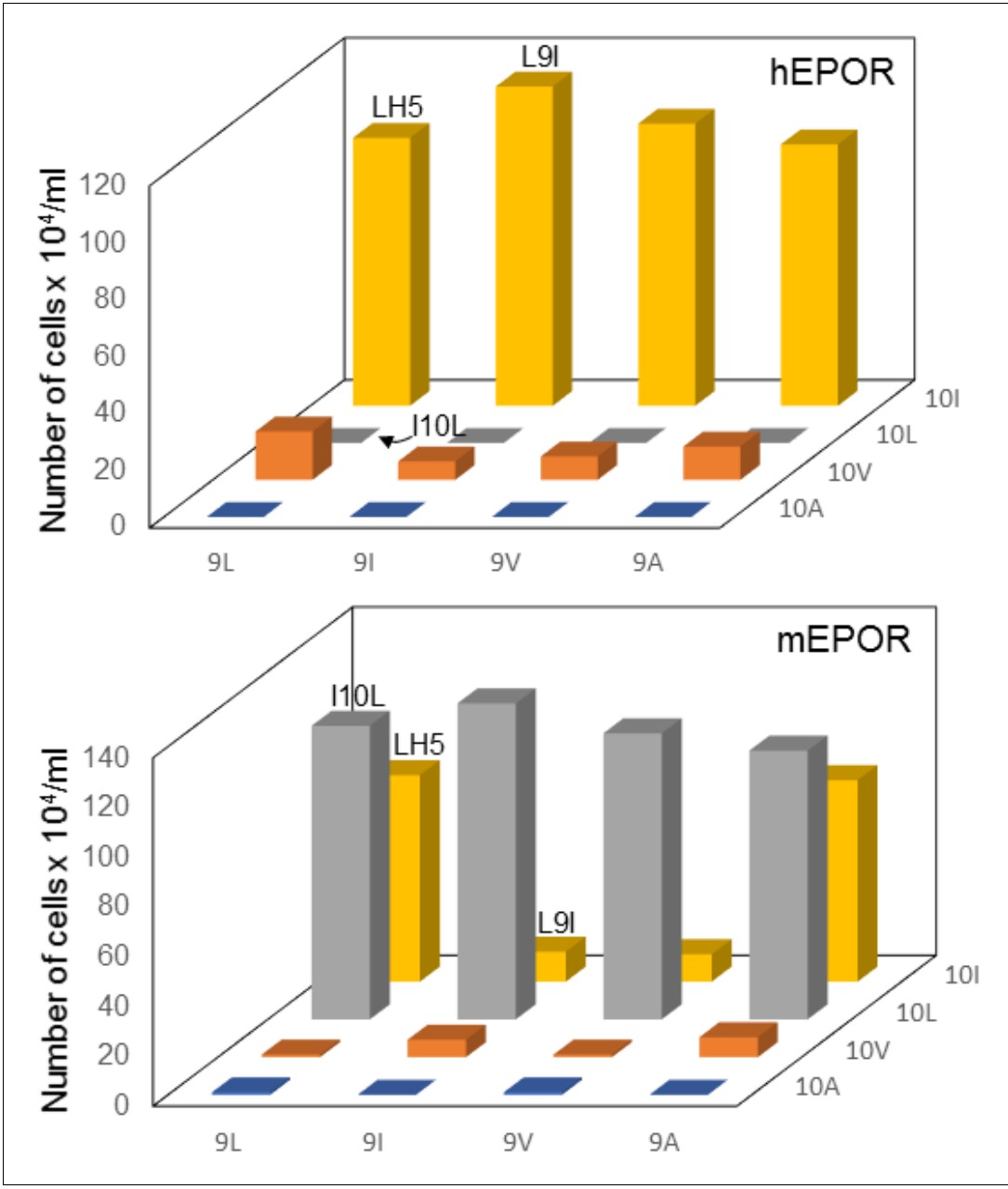

**Figure 5.** Mutational analysis of traptamer at positions 9 and 10. BaF3 cells expressing the hEPOR (upper panel) or the mEPOR (lower panel) were infected with MSCVp expressing wild-type LH5 or mutants containing all combinations of isoleucine, leucine, valine, and alanine at positions 9 and 10. After puromycin selection, cells were tested for IL-3 independence. The right axis indicates the residue at position 10. The horizontal axis indicates the residue at position 9. The vertical axis shows the number of live cells six days after IL-3 removal. The bars representing cells expressing LH5, LH5-L9I, and LH5-I10L are labeled. The bars show mean ± SEM for three independent experiments. The results for the LH5 sample were significantly different (p≤0.005, as assessed by two-tailed Student's t-test) compared to cells expressing each mutant, except for the mutants with isoleucine at position 10 (top panel) or leucine at position 10 and LH5-L9A (bottom panel).
DOI: https://doi.org/10.7554/eLife.27701.013

based on the hEPOR sequence) (*Figure 6A*). To identify the residues in the TMD of the mEPOR that dictate the specificity of the traptamers, we tested the effect of all three mouse-to-human single substitution mutations in the TMD of mEPOR. Cells expressing these mutants did not proliferate in

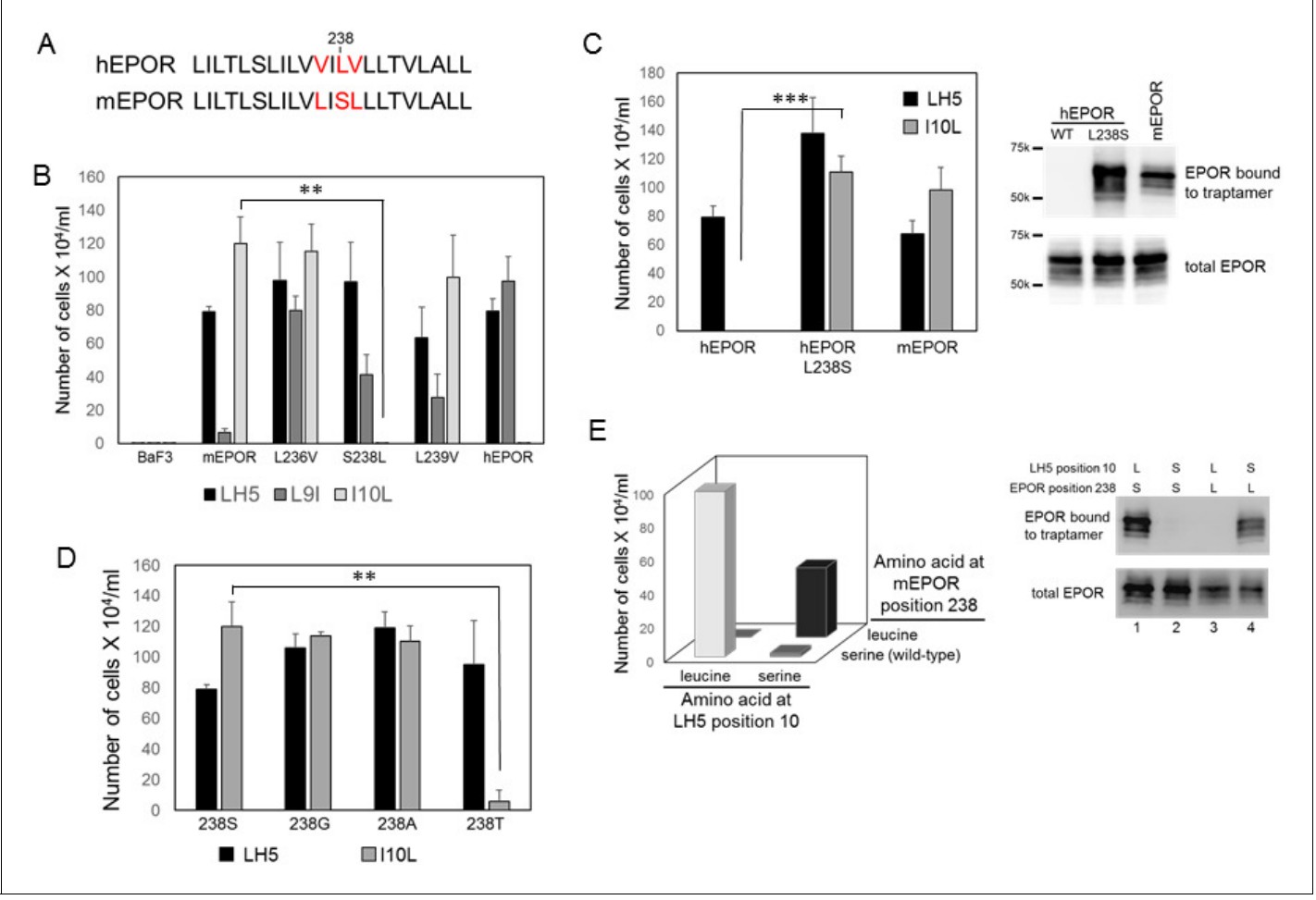

**Figure 6.** Identification of amino acids that mediate complex formation between the traptamers and the EPOR. (A) Sequence comparison of hEPOR and mEPOR TMD (residue 226–247). Sequence differences between mEPOR and hEPOR are shown in red. (B) BaF3 cells or BaF3 cells expressing the wild-type mEPOR, the wild-type hEPOR, or the mEPOR containing the indicated mutation were infected with retroviruses expressing the indicated traptamer. After puromycin selection, the cells were tested for IL-3 independence. The number of live cells was counted six days after IL-3 removal. The bars show mean ± SEM for three independent experiments. Statistical significance of differences between cells expressing wild-type mEPOR and cells expressing a mutant mEPOR were evaluated by two-tailed Student's t-test with unequal variance (**p≤0.02). (C) **Left panel**. BaF3 cells expressing wild-type hEPOR, L238S hEPOR mutant, or mEPOR were infected with retrovirus expressing FLAG-tagged LH5 or LH5-I10L. After puromycin selection, cells were tested for IL-3 independence. The number of live cells was counted six days after IL-3 removal is shown. The bars show mean ± SEM for three independent experiments. Statistical significance of differences between cells expressing wild-type hEPOR and cells expressing hEPOR/L238S were evaluated by two-tailed Student's t-test with unequal variance (***p≤0.005). **Right panel**. Extracts prepared from cells in left panel were immunoprecipitated with anti-FLAG antibody and immunoblotted with anti-HA antibody to detect EPOR associated with traptamer (top panel), or directly immunoblotted with anti-HA antibody to detect total EPOR (bottom panel). (D) BaF3 cells expressing wild-type mEPOR (238S) or the indicated mEPOR mutant were infected with retrovirus expressing LH5 or LH5-I10L. After puromycin selection, IL-3 independence was tested. The number of live cells was counted six days after IL-3 removal. The bars show mean ± SEM for three independent experiments. Statistical significance of differences between cells expressing wild-type mEPOR and cells expressing an mEPOR mutant were evaluated by two-tailed Student's t-test with unequal variance (**p≤0.02). (E) **Left panel**. BaF3 cells co-expressing FLAG-tagged LH5-I10L or LH5-I10S (horizontal axis) and the wild-type mEPOR or the S238L mEPOR mutant (right axis) were tested for IL-3-independence. The vertical axis shows the number of live cells six days after IL-3 removal. The parental combination (mEPOR/LH5-I10L) is shown in light grey bar; the compensating combination (mEPOR S238L/LH5-I10S) is shown in black bar. The bars show mean ± SEM for three independent experiments. Statistical significance of differences between cells co-expressing both mutants and cells expressing either single mutant was p≤0.05 as evaluated by two-tailed Student's t-test with unequal variance. **Right panel**. Extracts prepared from cells in left panel were immunoprecipitated with anti-FLAG antibody and immunoblotted with anti-HA antibody to detect EPOR associated with traptamer (top panel), or directly immunoblotted with anti-HA antibody to detect total EPOR (bottom panel).

DOI: https://doi.org/10.7554/eLife.27701.014

IL-3-free medium, demonstrating that they were not constitutively active (data not shown). As expected, LH5 cooperated with all three mutants (*Figure 6B*).

We determined whether these mutations affected the activity of the traptamer mutants. The activity of LH5-I10L was not affected by the mutation at position 236 or 239 in the mEPOR (*Figure 6B*). In contrast, the S238L mutation completely abrogated the ability of the mEPOR to respond to LH5-I10L, showing that Ser238 was important for specific recognition of the mEPOR by LH5-I10L (*Figure 6B*). To confirm the role of Ser238 in determining traptamer specificity, we mutated Leu238 in the hEPOR to serine and tested the response of this mutant to LH5-I10L. As shown in *Figure 6C*, left panel, LH5-I10L induced IL-3 independence in cells expressing the L238S hEPOR but not wild-type hEPOR. In addition, LH5-I10L formed a stable complex with the L238S hEPOR mutant but not with wild-type hEPOR (*Figure 6C*, right panel). Thus, the absence of serine at position 238 in the TMD of hEPOR is the sole reason why hEPOR does not respond to LH5-I10L.

The analysis of LH5-L9I activity with these mutants was not straightforward. The L236V mEPOR mutant displayed the greatest ability to cooperate with LH5-L9I, but the S238L and L239V mutants displayed moderate ability to support LH5-L9I-mediated IL3-independent growth (*Figure 6B*). Thus, several residues in the EPOR TMD controlled the response to LH5-I9L. Because of the secondary importance of position nine in determining traptamer specificity (*Figure 5*), we did not pursue the analysis of this position further.

We next determined which structural feature of Ser238 was required for activity with LH5-I10L. We mutated Ser238 to either of two small aliphatic residues, alanine and glycine, or to the polar residue threonine, which contains a side chain hydroxyl group like serine but is one methyl group larger. None of the mutants constitutively induced IL-3 independence (data not shown), and all were activated by LH5, as shown in *Figure 6D*. Traptamer LH5-I10L induced IL-3 independence with the wild-type mEPOR and with the S238A and S238G mEPOR mutants. Thus, alanine or glycine can replace serine at position 238, demonstrating that the serine hydroxyl group is not important for activity in this assay. In contrast, LH5-I10L did not cooperate with the S238T mEPOR mutant, suggesting that the size of the side chain at position 238, rather than hydrophilicity, determined the ability of LH5-I10L to productively interact with the mEPOR.

## Compensatory mutations within the TMD of the EPOR receptor and a traptamer

The results described above raised the possibility that the long side chain at position 10 in the traptamer directly interacted with the short side chain of mEPOR TMD at position 238. If the residues at traptamer position 10 and mEPOR position 238 interact directly, we hypothesized that simultaneously swapping the residues at both of these positions might restore biological activity. We first tested the activity of a LH5 mutant with serine at position 10. As shown in *Figure 6E* (left panel), LH5-I10S did not cooperate with wild-type mEPOR to induce IL-3 independence. Strikingly, however, LH5-I10S cooperated with mEPOR/S238L to restore a significant level of IL3-independence (*Figure 6E*, left panel, black bar). Thus, LH5-I10S did not cooperate with the wild-type mEPOR and LH5-I10L did not cooperate with mEPOR S238L, but LH5-I10S did cooperate with mEPOR S238L.

We also conducted co-immunoprecipitation experiments for the four combinations of proteins. As shown in *Figure 6E* (right panel), complex formation between mEPOR and LH5-I10L was abrogated by mutating position 10 in the traptamer to serine (lane 2) or by mutating Ser238 to leucine in the TMD of the mEPOR (lane 3). Notably, simultaneous mutation of position 10 of the traptamer and position 238 of the mEPOR restored complex formation (lane 4), consistent with restoration of biological activity by this combination of mutations. Restoration of activity by compensating mutations also implies that the defect of the position S238L mEPOR mutant is caused by direct disruption of the interaction with the traptamer and not, for example, by an indirect effect of the mutation on the receptor itself. Taken together, these results provide strong genetic evidence that the two proteins contact each other directly and suggest that this interaction is mediated by contacts between the side chains at position 10 in the traptamer and position 238 in the TMD of the mEPOR.

## Traptamers remodel the conformational landscape of EPOR TMD dimers in simulations

The TMD of the mEPOR (and to a lesser extent, the hEPOR) has intrinsic ability to homodimerize (*Ebie and Fleming, 2007*; *Kubatzky et al., 2001*; *Ruan et al., 2004*; *Li et al., 2015*). In addition to dimerization, the conformational state of the dimeric EPOR TMDs regulates activity (*Moraga et al., 2015*; *Seubert et al., 2003*). We thus used molecular dynamics (MD) simulations in an implicit membrane to determine whether traptamers may affect the conformation of EPOR TMDs within the dimer. Innate conformational preferences of mEPOR and hEPOR TMD dimers were first established by simulating these in the absence of traptamers (total simulation time of 1 μs; see Materials and methods). Ten independent simulations were run for each receptor, and inter-helical geometries in resulting trajectories were analyzed, focusing on helical phase. This parameter describes the rotation of individual TM helices about their helical axes, thus defining which helical faces pack into the dimer interface (*Grigoryan and Degrado, 2011*). Prior analysis (*Seubert et al., 2003*) implies that close-to-symmetric configurations are likely most relevant in the context of the full-length EPOR, so we concentrated our analysis on these states. The helical phase distributions

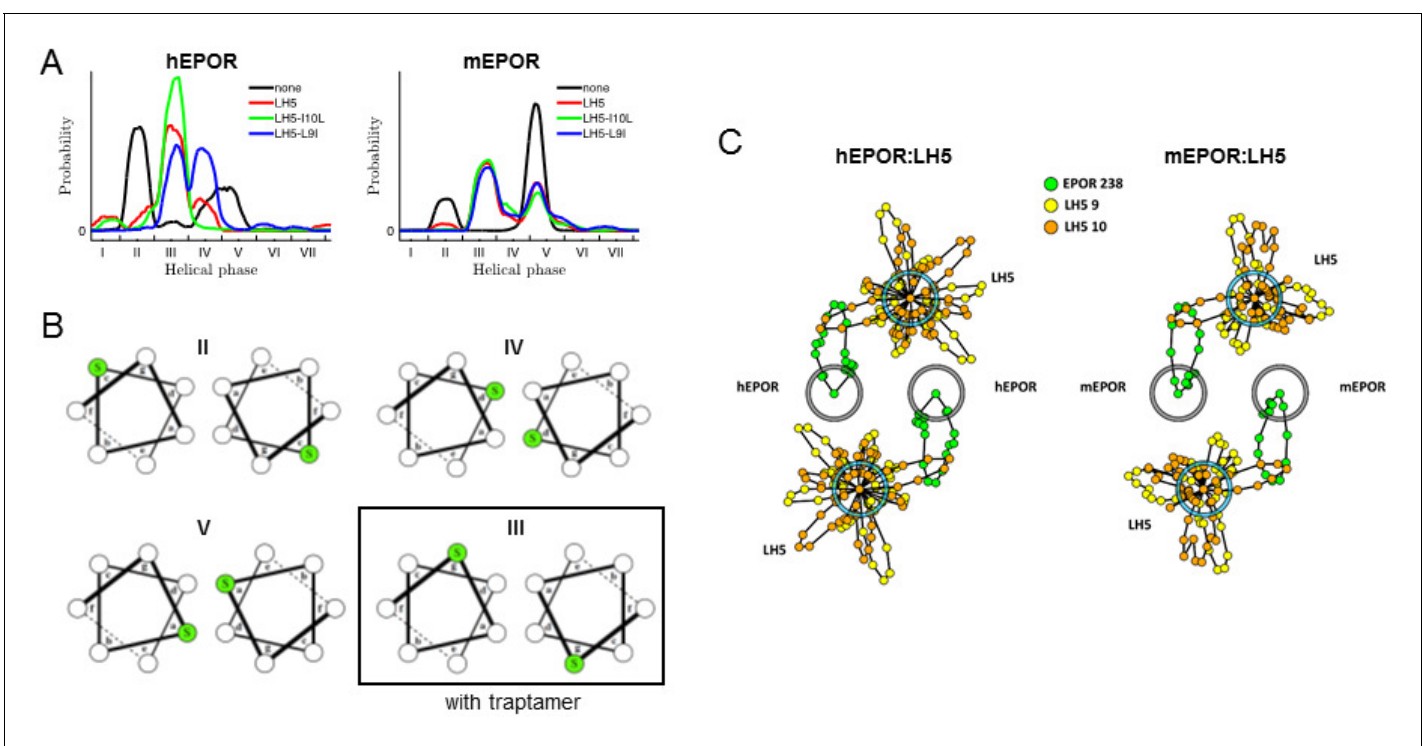

**Figure 7.** Molecular modeling of the effect of the traptamers on the EPOR TMD. (A) Helical phase distributions of hEPOR (left) and mEPOR (right) TMD dimers in the absence of traptamer (black lines) or in 2:2 stoichiometry with LH5 (red lines), LH5-L9I (blue lines) or LH5-I10L (green lines). The range of the x-axis is from −180° to 180°, with labeled ticks denoting phase boundaries corresponding to canonical coiled-coil heptad registers, labeled I through VII. (B) Helical wheel diagrams of mEPOR TMD dimers corresponding to the basal states II, IV, and V, as well as state III, which is appreciably populated only in the presence of traptamers. Ser238 is colored green. (C) A diagram of state III (gray and blue circles represent EPOR TMDs and traptamers, respectively), showing the helical phase distributions from simulations of hEPOR (left panel) and mEPOR (right panel) TMD dimers in complex with LH5. Phase distributions are plotted in polar coordinates (rosette plots): each dot representing a bin in the histogram, with its location around the circle indicating the phase value and its distance from the helix center denoting the frequency of the bin (points farther from the center are more frequent).

DOI: https://doi.org/10.7554/eLife.27701.015

The following figure supplements are available for figure 7:

**Figure supplement 1.** Helical wheel diagrams and helical phase distributions of mEPOR TMD dimers.

DOI: https://doi.org/10.7554/eLife.27701.016

**Figure supplement 2.** Time to approach symmetry.

DOI: https://doi.org/10.7554/eLife.27701.017

within such pseudo-symmetric conformations for both the hEPOR and mEPOR TMD are shown in *Figure 7A* (black curves), with states corresponding to canonical coiled-coil heptad registers (*Grigoryan and Degrado, 2011*) labeled with Roman numerals (see *Figure 7B* and *Figure 7—figure supplement 1* for corresponding helical wheel diagrams). Two states dominated both hEPOR and mEPOR: one corresponding to register **II** and the second involving registers **IV** and **V**. These states placed polar serine and threonine side chains in the dimer interface, enabling interhelical hydrogen bonds. Published work suggests that these EPOR TMD orientations do not result in robust downstream signaling (*Seubert et al., 2003*).

We next asked if these state distributions were perturbed by the traptamers. We considered ratios of one, two, or four traptamer molecules per EPOR TMD dimer (designated 1:2, 2:2, and 4:2, respectively), with traptamers inserted in the membrane anti-parallel to EPOR TMDs. In addition, traptamers were restrained to remain in the proximity of the dimeric EPOR TMDs. *Figure 7A* shows the helical phase distributions for EPOR TMD dimers in the presence of LH5 (red line), LH5-L9I (blue line), or LH5-I10L (green line) in the 2:2 stoichiometry. Introduction of traptamers caused substantial remodeling of the conformational landscape of the EPOR TMD dimers, as it did in the other stoichiometries we tested (see *Figure 7—figure supplement 1B*). Notably, state **III** was nearly unpopulated with EPOR TMD dimers alone, but was sampled robustly in the presence of the traptamers. In state **III**, residue 238 of EPOR TMD mapped to a **g** position, making it available for quaternary interactions with traptamers (*Figure 7B*; Ser238 labeled green), consistent with the experimentally-observed importance of position 238 in traptamer action. In contrast, in basal states **IV** and **V**, residue 238 was buried in the TMD dimer interface and therefore not accessible to the traptamer(s) and in basal state **II**, residue 238 was in position **c** far from the EPOR dimer interface. Remarkably, the placement of a single side chain methyl group at traptamer position 9 or 10 affected the phase distributions. For example, state **IV** in the hEPOR was populated in response to LH5 and LH5-L9I, but not in response to LH5-I10L. However, in the modeling there was no evident correlation between the biological activity of the wild-type and mutant traptamers against the different species of the EPOR and the ability of the traptamers to induce specific orientations of the EPOR TMDs. This is presumably because the biological activity of a particular mutant/EPOR combination arises from the ability of the traptamer to bind the EPOR, induce EPOR dimerization, and to trap it in a functional orientation, whereas our simulations capture only the latter effect because the sampling was limited to the associated state only.

To better understand how traptamers remodel the state distribution of EPOR TMDs, we considered state **III**, which is most clearly modulated in response to traptamers. We identified all snapshots from all trajectories that map to state **III** and extracted helical phases corresponding to position 238 in the EPOR TMD and positions 9 and 10 in the traptamers; these are represented as rosette plots in *Figure 7C* for hEPOR/LH5 and mEPOR/LH5 complexes. This analysis shows that positions 9 and 10 spend much of their time oriented towards the EPOR TMD, in a phase orientation that could enable them to engage EPOR position 238. In fact, analysis of simulation trajectories revealed that positions 9 or 10 are within van der Waals contact of position 238 (for at least one of the TMD helices) in 55% to 97% of all state-**III** trajectory snapshots, across all traptamers in all three tested stoichiometries. This is consistent with our genetic findings that TMD position 238 and traptamer positions 9 and 10 play crucial roles in establishing traptamer specificity in cells. *Figure 7C* also shows that there is considerable heterogeneity in traptamer rotational state, which suggests that traptamers act in a much more *dynamic* fashion than may typically be expected of signaling modulators. That is, rather than being frozen in a single preferred configuration with respect to the EPOR TMD dimer, traptamers appear to adopt a variety of orientations, yet still making specific contacts with the EPOR TMD and having a net effect of perturbing its conformational equilibrium.

NMR data suggest that EPOR TM dimers are symmetric, with or without traptamers, whereas we observe that many individual snapshots sampled in MD trajectories are significantly non-symmetric. Indeed, MD simulations showed that traptamers rapidly inter-convert between different helical phases in all stoichiometries, such that complexes can be expected to be symmetric only in an average sense. In fact, analysis of our MD trajectories revealed that all simulations are consistent with complexes approaching symmetry much faster than the typical ms NMR time scales (see *Figure 7—figure supplement 2*).

Overall, these simulations demonstrated that traptamers significantly remodel the conformational landscapes of EPOR TMD dimers, shifting the populations towards states in which TMD position 238

interacted with traptamer residues 9 and 10. Furthermore, subtle chemical differences between trap-tamers can have dramatic consequences on this orientation. We point out that we have not estab-lished which EPOR TMD state is most relevant for EPOR activation by the traptamers, what the correct EPOR-traptamer stoichiometry is or, indeed, that there even is a single well-defined stoichi-ometry. Nevertheless, our results strongly suggest that the active LIL traptamers exhibit their func-tional effects through structurally specific interactions with the EPOR TMDs, even though they are likely to be dynamic and adopt a variety of conformations in complex with their target TMDs, consis-tent with their low information-content sequences.

## All-atom simulations of the EPOR/traptamer complexes

Because the impact of lipids and water on interactions between the traptamers and the EPOR TMDs was not captured by the implicit modeling, we performed explicit-solvent all-atom MD simulations of the active and mutant traptamer-EPOR systems in 1-palmitoyl-2-oleoyl-sn-glycero-3-phosphocho-line (POPC) bilayers. Representative 2:2 centroid models obtained from the implicit membrane-based simulations were used as starting points for all-atom simulations after adding short juxtamem-brane domains to each end of the TMDs of the receptors. The helices were not restrained to remain associated in this modeling. After equilibration of the protein-bilayer system for 7 ns, each simulation was carried out for 50 ns (production run) in Gromacs 2016.1 in duplicate, with different random seeds. Although the resulting trajectories are not long enough to observe significant inter-state con-versions of EPOR TMD dimers as with implicit-solvent simulations, several features of EPOR/trap-tamer complex were evident.

The EPOR TMDs remained associated with each other and with traptamers for the duration of the simulation in all systems and all replicates. Backbone root-mean-square-deviation (RMSD) of the hEPOR TMD, relative to the equilibrated conformation, increased significantly in the presence of the parental LH5 traptamer, underscoring the dynamic nature of the complex, whereas the mEPOR TMD was much less dynamic than the hEPOR (*Figure 8A*). When the root mean square fluctuation (RMSF) of individual residues were compared, these striking differences in the dynamics of the hEPOR and mEPOR TMDs in the presence of LH5 were distributed along the entire length of the TMDs in both complexes (*Figure 8B*). Compared to LH5, the mutant traptamers caused a marked reduction in the dynamics of the hEPOR TMD along the length of the hEPOR TMD (*Figure 8A*, top, and *Figure 8C*). In contrast, dynamics of the mEPOR TMD were similar in the presence of LH5 and its mutants (*Figure 8A*). All of these trends were observed in both independent simulation replicas performed in each system. This analysis showed that the hEPOR and mEPOR TMDs responded differently to the traptamers and that repositioning a single methyl group in a traptamer can markedly affect the dynamics of the hEPOR TMD dimer along its entire length.

## Discussion

By studying simple model TM proteins, we discovered that interactions between TMDs can display high specificity in mammalian cells that can be precisely modulated by minimal structural differences in the interacting helices. The small size and chemical simplicity of the traptamers allowed us to ana-lyze the basis for specificity in the absence of confounding factors such as contributions from protein segments outside of the membrane or hydrogen bonding between hydrophilic amino acid side chains. We discovered that single leucine-isoleucine substitutions at two positions in a traptamer, which change the placement of single methyl groups, determined whether it bound and activated the hEPOR, the mEPOR, or both receptors. Furthermore, corresponding changes on the receptor rescued activity. Because of the low sequence complexity of the traptamers and the minimal chemi-cal differences that confer specificity, their ability to distinguish between the hEPOR and mEPOR is intriguing and potentially implies that entirely novel design principles can modulate the activity of TM proteins.

How does a protein composed of a 25-residue sequence of two similar hydrophobic amino acids activate a specific cellular target and discriminate between two closely related targets based on the placement of a single side chain methyl group in the sequence? Our data show that traptamer action is mediated by direct binding to the EPOR TMD. The biological activity of LH5 and its mutants corre-lates perfectly with their ability to associate with the hEPOR or mEPOR, the ability of the traptamers to activate the EPOR is determined by the sequence of the EPOR TMD, and the isolation of

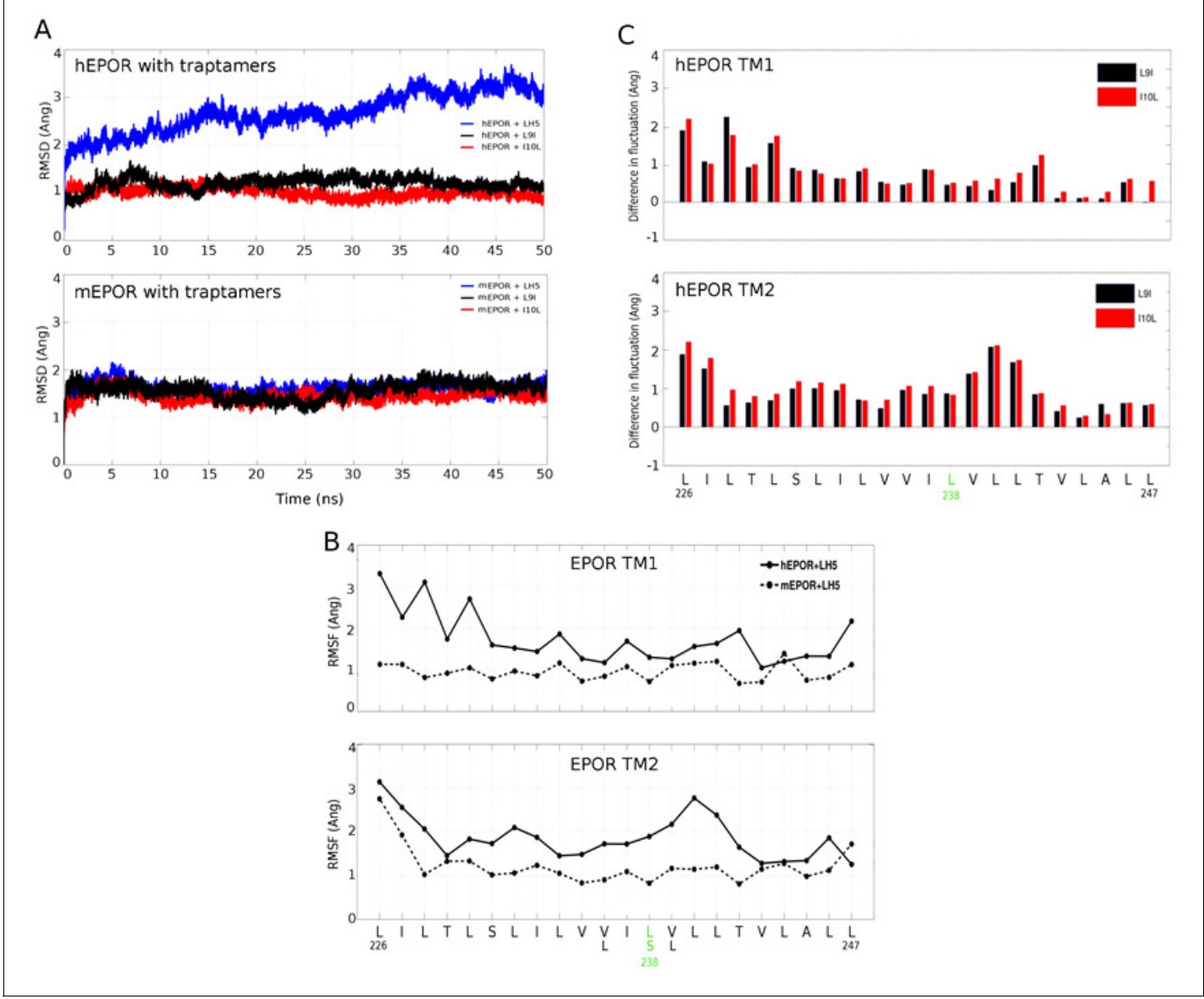

**Figure 8.** All-atom simulations of the EPOR/traptamer complexes. (**A**) RMSD profiles for the EPOR TMDs for the various receptor and traptamer systems. RMSD values for hEPOR (top panel) and mEPOR (bottom panel) over time in the presence of LH5 (blue line), LH5-L9I (black), and LH5-I10L (red) were computed for the backbone atoms within the EPOR TMD with respect to their starting structures. (**B**) Per-residue RMSF profiles of the hEPOR (solid line) and mEPOR (dashed line) TMDs in the presence of LH5 over 50 ns of production run. Comparisons for each TMD in the dimer are depicted in separate panels. The membrane-spanning sequence of hEPOR (L226-L247) is shown below the bottom panel, with the residues differing in the mEPOR shown below it. Leu238 and Ser238 in hEPOR and mEPOR, respectively, are highlighted in green. (**C**) Difference in per-residue RMSF for the TMDs of the hEPOR in presence of the mutant traptamers relative to the hEPOR-LH5 traptamer complex (computed over 50 ns of production run). Differences between LH5 and LH5-L9I and between LH5 and LH5-I10L are plotted as black and red bars, respectively, and depicted in separate panels for each hEPOR TMD. The membrane-spanning sequence of the hEPOR (L226-L247) is shown below the bottom panel with Leu238 highlighted in green.

DOI: https://doi.org/10.7554/eLife.27701.018

compensatory mutations in the traptamer and the mEPOR TMD constitutes formal genetic proof of a direct interaction. Finally, the molecular modeling and NMR studies provided additional evidence that the traptamers specifically interact with the EPOR TMD in simplified systems lacking other proteins. We therefore conclude that direct binding of the traptamers to the EPOR is mediated by complementarity between hydrophobic surfaces generated by the sequence of leucines and isoleucines

in the traptamer and the sequence of the EPOR TMD. There are more than 33 million ($2^{25}$) possible 25-residue sequences of leucine and isoleucine, suggesting that even such short, simple proteins can generate a large number of surfaces and thus potentially interact with a wide variety of target proteins.

The discovery of LH5 mutants that activate only the hEPOR or the mEPOR eliminates many trivial explanations for the specificity of the traptamers. Thus, the inability of such mutants to activate a particular EPOR is not the result of poor expression, incorrect localization, gross misfolding or major structural rearrangement or aggregation of the mutant traptamer, or of differential partitioning of the membrane into domains resulting in non-specific EPOR clustering and activation. Rather, our results strongly argue that the traptamers act by making specific, yet dynamic, contacts with the EPOR TMDs and that the amino acids at positions 9 and 10 in the traptamer and at specific positions in the EPOR TMD play key roles in this interaction, ultimately manifesting in differential ability to activate the human or mouse EPOR. Because van der Waals interactions require precise geometry and spacing, moving a methyl group at position 9 or 10 in LH5 one carbon closer or farther from the polypeptide backbone, respectively, generates a novel surface that is now complementary to only the hEPOR or the mEPOR, thereby altering specificity. Thus, the methyl group at these positions is acting as a molecular toggle switch to determine traptamer specificity. The ability of a traptamer to toggle back and forth between closely related targets in response to the placement of a single methyl group implies that the determinants of specificity within a membrane are controlled by minimal chemical differences.

The addition of the traptamers did not result in additional significant changes in the backbone NMR chemical shifts besides those resulting from dimerization. This observation suggests that the LIL traptamers have lipid-like chemical properties because of their low-complexity sequence containing purely hydrocarbon side chains, a feature which may favor dynamics in binding and allow for diversity in stoichiometry. It is therefore possible that traptamer:TMD complexes form un-conventional dynamic complexes, similar to the fuzzy interactions recently described for intrinsically disordered proteins (*Klein et al., 2003*; *Mittag et al., 2008*; *Olsen et al., 2017*), which may nevertheless still be of high affinity. The ability of the traptamers to productively interact in a highly specific manner with particular EPOR TMDs suggests that such dynamic interactions can also display high specificity in membranes. A route by which the traptamers would avoid non-specific interactions might be through a competition between the association of hetero- and homo-oligomeric complexes of EPOR as well as of the traptamer. The lipid-like properties of the traptamers may lead to the formation of traptamer polymer-islands in the membrane, implying that in order to bind a certain receptor, the traptamer:receptor interaction needs to be stronger than the internal affinity between traptamers and between the EPOR TMDs dimers in their inactive state.

Although the oligomerization state of the traptamer itself is not known, our co-immunoprecipitation and NMR experiments indicate that traptamer binding drives the EPOR toward the oligomeric, likely dimeric, form. Signaling by the EPOR is dependent on the orientation of its TMDs and linked cytoplasmic domains (*Moraga et al., 2015*; *Seubert et al., 2003*; *Syed et al., 1998*; *Remy et al., 1999*). In the absence of ligand, the TMDs of the mEPOR are in an inactive orientation stabilized by hydrogen bonds across the homodimer interface ([*Ruan et al., 2004*; *Seubert et al., 2003*; *Li et al., 2015*], and our MD results), but when the mEPOR dimer is in an active conformation Ser238 appears to be oriented away from this interface (*Seubert et al., 2003*). This implies that the traptamers not only stabilize the EPOR dimer but also cause it to adopt a productive orientation. Consistent with this interpretation, our implicit-membrane MD simulations indicate that the traptamers induce a fraction of the mEPOR TMDs to re-orient from an inactive orientation into one where Ser238 engages the traptamer *via* van der Waals interactions. Because of uncertainties in the stoichiometry of the complex, among other parameters, we cannot identify with certainty the active orientation, but rather our results imply that binding of the traptamers is likely to reorient the EPOR TMD from an inactive orientation into an active one. The differential response of the mEPOR and the hEPOR to the traptamers in the all-atom MD simulations raises the possibility that there are differences in the mechanisms by which the traptamers activate the two species of the EPOR.

The small size and chemical simplicity of the traptamers allowed us to interrogate all possible leucine-isoleucine substitutions in LH5. The results of these experiments established that specificity is determined by repositioning the methyl group at only a limited number of positions in LH5, whereas mutations that inhibit activity against both EPORs are distributed throughout the sequence. These

results imply that the interaction of LH5 with the EPOR TMD is stabilized by multiple weak van der Waals interactions along the length of the TM helices, which, in aggregate, are sufficient to overcome the interhelical hydrogen bonds that stabilize the inactive conformations of the EPOR TMD. In such an association driven by shape complementarity, moving a single methyl group at numerous positions can cause steric clashes that disrupt or weaken the binding of the traptamer to the TMD of both the hEPOR and the mEPOR. In contrast, mutations at a few positions fine-tune specificity by generating surfaces that bind strongly to the TMD of only the hEPOR or the mEPOR, while retaining the ability to induce EPOR TMD dimerization and reorientation. Interestingly, the all-atom simulations demonstrated that the single leucine-isoleucine substitutions in the mutant traptamers generate conformational differences along the length of the hEPOR TMDs, showing that the structural consequences of these seemingly simple changes can be complex. We note that substitutions at positions 2 and 3 in LH5 also resulted in preferential activation of the hEPOR and the mEPOR, respectively, although the differences were not as dramatic as those caused by the L9I and I10L mutations (*Figure 2C*). Because every seventh residue is on the same face of a TM coiled-coil, positions 2 and 9 are on one face of the TMD helix, and positions 3 and 10 are on a different face, implying that these two faces play particularly important roles in determining specificity.

Detailed analysis of a single traptamer does not provide a complete picture of the TM interactions that can drive EPOR activation. The active traptamers fall into two classes. LH1 to LH4 activate only the hEPOR, and LH5 activates both hEPOR and mEPOR. As shown in the boxes in *Figure 1B*, LH5 differs from the other traptamers at eleven positions, including the major specificity-determining positions 9 and 10. These striking sequence and specificity differences suggest that these two classes of traptamers interact with the hEPOR in fundamentally different ways, presumably though the formation of different interaction surfaces or different oligomeric states.

By developing a genetic system to study small TM proteins consisting of only leucines and isoleucines, we discovered that highly specific interactions can be fine-tuned by extremely subtle chemical differences confined to the membrane spanning segment and that this specificity can be driven exclusively by van der Waals interactions. The traptamers were not subjected to natural selection to eliminate undesired activities or interactions, implying that this high specificity is an intrinsic property of TMDs in the context of membrane lipids. This may be particularly true for TMDs that consist exclusively of hydrophobic residues, whose interaction partner(s) is dictated solely by shape complementarity along the length of the interacting helices. In contrast, hydrophilic amino acids can form interhelical hydrogen bonds or ionic interactions, in which interacting side chains may be sufficient to drive alternative TMD associations (*Choma et al., 2000*; *Zhou et al., 2000*).

Although we studied extremely simple model transmembrane proteins, the principles uncovered here are likely to apply to at least some naturally occurring TMDs. For example, the TMD of the BPV E5 protein contains a stretch of 12 consecutive residues comprised exclusively of leucine and phenylalanine, and the TMD of the human colony stimulating factor receptor contains a conserved homopolymeric stretch of 10 consecutive leucines. In addition, the TMD of the PDGF β receptor contains diverse amino acids, but a leucine to isoleucine mutation in the middle of this TMD prevents binding of the E5 protein (*Edwards et al., 2013*). Our results imply that naturally occurring TMD interactions based on shape complementarity would also display very high specificity, unless there was an evolutionary advantage to relax this specificity. During evolution, incorporation of a limited number of hydrophilic amino acids into TMDs consisting of otherwise hydrophobic amino acids may allow homologous interactions with a family of related but not identical TMDs containing hydrophilic residues, or may generate higher affinity interactions, even if that comes at the cost of reduced specificity. Our findings also show that supposedly conservative polymorphisms or mutations in TMDs can have profound effects on TM protein function and should be strictly evaluated for this possibility. These features of TMD interactions may also inform the design of biologically active TM proteins. Finally, our results suggest that agents including small hydrophobic molecules and peptides that disrupt or modulate specific interactions between TMDs could have valuable research and applied uses. Genetic selection of traptamers will be a powerful tool to elucidate additional rules governing transmembrane interactions and to identify lead compounds for these approaches.

## Materials and methods

### Vectors and cloning

The HA-tagged hEPOR and HA-tagged mEPOR genes (originally obtained from S. Constantinescu, Ludwig Institute) were excised from the pBABE-puro retroviral vector and subcloned into pMSCV-neo (Clontech, Mountain View, CA) using EcoRI and HpaI restriction sites. An MluI restriction site was introduced into the mEPOR and hEPOR genes as a silent mutation at position 642 (ACG CG*C* to ACG CG*T*) by using standard QuickChange Site-Directed Mutagenesis protocol and *Pfu* Turbo polymerase (Agilent Technologies, Santa Clara, CA). The mhm chimeric EPOR and mEPOR mutants containing point mutations in the mEPOR TMD were constructed by cloning double-stranded DNA gBlock Gene Fragments (Integrated DNA Technologies, Redwood City, CA) containing the wild-type hEPOR TMD or the desired mutation(s) in the mEPOR TMD into mEPOR gene using MluI and ApaI restriction sites (corresponding to amino acids Arg214 to Pro328 of the mEPOR protein). The hmh chimeric EPOR and the L238S mutant of the hEPOR were constructed by first cloning double-stranded DNA gBlock Gene Fragments containing the wild-type mEPOR TMD or the L238S mutation in the hEPOR TMD into the hEPOR gene in pBABE-puro vector using MluI and BglII restriction sites (corresponding to amino acids Arg215 to Lys256 of the hEPOR protein), and then sub-cloned into pMSCV-neo using EcoRI and HpaI restriction sites. The Δ259 hEPOR truncation mutant was constructed in pMSCV-neo by using Phusion High Fidelity DNA polymerase(ThermoFisher Scientific, Waltham, MA) to delete the codons between codon 259 and the stop codon. The F8 mutant of the hEPOR was constructed by replacing sequences encoding the cytoplasmic segment of the HA-hEPOR gene with a DNA gBlock Gene Fragment containing eight tyrosine-to-phenylalanine mutations. Mutants of traptamer LH5 were constructed by cloning double-stranded DNA gBlock Gene Fragments containing the desired sequence into pMSCV-puro using XhoI and EcoRI restriction sites.

### Cells, retrovirus infections, and activity testing

Retrovirus stocks were prepared in human embryonic kidney (HEK) 293 T cells, which were obtained from the American Type Culture Collection (CRL3216, Manassas, VA) and maintained in DMEM-10 medium: Dulbecco's Modified Eagle Medium (DMEM) supplemented with 10% fetal bovine serum (FBS) (Gemini Bioproducts, West Sacramento, CA), 4 mM L-glutamine, 20 mM HEPES (pH 7.3), and 1X penicillin/streptomycin (P-S). All biological experiments were performed in murine interleukin-3 (IL-3)-dependent BaF3 cells, which were obtained from Alan D'Andrea (Dana-Farber Cancer Institute) and maintained in RPMI-10 media: RPMI-1640 supplemented with 10% heat-inactivated FBS, 5% WEHI-3B cell-conditioned medium (as the source of IL-3), 4 mM L-glutamine, 0.06 mM β-mercaptoethanol, and 1X P-S. We used several phenotypic and biochemical markers to authenticate our cell lines. BaF3 cells have a distinctive morphology, and unlike the other cells used in the laboratory they grow in suspension. The presence or absence of expression of EPOR and traptamers in the appropriate cell line was confirmed by western blotting (e.g. see *Figure 3D* and *Figure 3—figure supplement 1*). We confirmed that all cells were IL-3 dependent unless they expressed the EPOR and an active traptamer. EPOR integrity was confirmed by showing that cells expressing EPOR could proliferate in the presence of EPO, whereas EPO did not support proliferation in cells lacking EPOR expression. These points are illustrated, for example, in *Figure 2A*. All cells were shown to be mycoplasma-free.

To produce retrovirus stocks, 293 T cells were transfected by mixing 2 μg pantropic pVSV-G (Clontech), 3 μg pCL-Eco (Imgenex, Littleton, CO), 5 μg of the retroviral expression plasmid of interest with 250 μl of 2x HEBS. 250 μl of 0.25 M calcium chloride was then bubbled into each mixture. The mixture (~500 μl) was incubated for 20 min at room temperature and then added drop-wise into $2.0 \times 10^6$ 293 T cells plated in 100 mm tissue culture dishes in DMEM-10. After the cells were incubated with the transfection mixture for 6–8 hr at 37°C, the medium was replaced with 5 mL fresh DMEM-10 medium. The cells were incubated for another 48 hr at 37°C, then the viral supernatant was harvested, filtered through a 0.45 μm filter (Millipore, Danvers, MA), and either used immediately or stored at −80°C.

For IL-3 independence assays, $2 \times 10^5$ BaF3 cells expressing the appropriate genes were washed in PBS three times to remove IL-3. Cell pellets were resuspended in 10 mL RPMI-10 IL-3-free medium, in which WEHI-3B cell-conditioned medium was not included. Viable cells were counted six

days after IL-3 removal. All IL-3 tests were performed in three independent biological replicates (i.e. independent infections to express traptamers followed by cell counts after IL-3 removal). All reported experiments included positive and negative controls that performed as expected, and no outliers in these experiments were excluded. All graphs show average values for IL-3 tests ± SEM. Statistical significance of differences between control and experimental samples was evaluated by two-tailed Student's t-tests with unequal variance, performed using T.TEST function in Microsoft Excel (2013). Significance cut-off for differences in cell numbers was established by p value < 0.05.

BaF3 cells expressing HA-tagged hEPOR, HA-tagged mEPOR, and all EPOR mutants were generated by infecting BaF3 cells with pMSCV-neo vector containing the desired EPOR gene. $5 \times 10^5$ BaF3 cells were washed with phosphate buffered saline (PBS) and then re-suspended in 500 µl RPMI-10 medium with 4 µg/mL polybrene. 500 µl retroviral supernatant or 500 µl DMEM-10 for mock-infection was added to re-suspended cells and incubated for 6 hr at 37°C. After incubation, 9 ml RPMI-10 was added and the BaF3 cells were incubated overnight at 37°C prior to selection in 1 mg/mL G418 or 1 µg/mL puromycin, as appropriate.

BaF3 cells co-expressing HA-tagged full-length hEPOR and truncated hEPOR (Δ259) were generated by infecting BaF3 cells as above with pMSCV-neo expressing Δ259, followed by selection with 1 mg/mL G418 for five days. The selected cells were then infected with pMSCV-neo containing the full-length hEPOR gene, followed by selection with 0.25 mg/mL G418 plus 0.6 U/ml human recombinant EPO for three days in the absence of IL-3. Immunoblotting with anti-HA antibody confirmed the expression of full-length and truncated EPORs. Traptamers cloned in MSCV-puro were introduced into these cells by infection followed by puromycin selection.

## Retroviral library construction, Screening, and Recovery

The UDv6 library was previously described (*Heim et al., 2015*). The library, containing sequences encoding an estimated five million different traptamers (based on the number of bacterial colonies pooled), was transfected into 293 T cells to generate a retrovirus stock. For the screening, six million BaF3/hEPOR cells were divided equally into six 25 cm² flasks with 1 ml RPMI-10 media in each flask and then infected with 1 ml of 20X concentrated retroviruses containing the UDv6 library at multiplicity of infection of ~1. Polybrene was added to a final concentration of 4 µg/mL.

Cells were incubated at 37°C for four hours before 8 ml fresh RPMI-10 media was added to each flask. 24 hr post-infection, 1 µg/mL puromycin was added to each flask. The puromycin selection continued for 3–4 days until all cells from mock infection were dead. $5 \times 10^5$ cells were then collected from each pool, washed twice in PBS, and resuspended in 10 mL medium containing 1/20 of the standard IL-3 concentration (reduced IL-3 medium). After 14 days, live cells were collected from each pool. Genomic DNA (gDNA) was isolated using DNeasy Blood and Tissue Kit ( Qiagen, The Netherlands). Library inserts were recovered from gDNA by using PCR with short primers specific to the library (as described in [*Heim et al., 2015*]). PCR products were then purified, digested with EcoRI and XhoI, and cloned into pMSCV-puro vector for the second round of screening.

DNA from the first round of screening was combined and transfected into 293 T cells to generate a secondary retrovirus stock. The second round screening was similar to the first round except that cells were resuspended in 10 mL medium without IL-3, rather than in reduced IL-3 medium. Gene fragments were recovered from the gDNA of cells proliferating in the absence of IL-3, cloned into pMSCV-puro vector, and sequenced.

## Immunoprecipitation and immunoblotting

Prior to collection, BaF3 cells were starved in RPMI-10 IL-3-free media for 3 hr at 37°C and, in some cases, acutely stimulated with 5 U/mL EPO for 10 min at 37°C. Cells were then washed twice with ice-cold PBS containing 1 mM phenylmethylsulfonyl fluoride (PMSF). For phosphotyrosine and phospho-protein blots, 1X HALT Protease and Phosphatase Inhibitor Cocktail (ThermoFIsher Scientific) and 500 µM hydrogen peroxide-activated sodium metavanadate were also added. Cells were lysed in FLAG-lysis buffer (50 mM Tris pH 7.4, 150 mM NaCl, 1 mM EDTA, 1% Triton-100) supplemented with protease and phosphatase inhibitors as above. All lysates were incubated on ice for 20 min, followed by centrifugation at 14,000 rpm for 30 min at 4°C. The total protein concentration of the

supernatants was determined using a bicinchoninic acid (BCA) protein assay kit (ThermoFisher Pierce, Waltham, MA).

To immunoprecipitate HA-tagged EPORs for phosphotyrosine blotting, 2 μl of anti-HA antibody (clone C29F4, Cell Signaling, Danvers, MA) was added to 0.5 mg of total protein and rotated overnight at 4°C. 50 μl Protein A Sepharose bead slurry was added and rotated for two hours at 4°C. To immunoprecipitate full-length hEPORs to assay receptor dimerization, 8 μl of a rabbit anti-EPOR polyclonal antibody (clone C-20, Santa Cruz Biotechnology, Dallas TX) was added to 0.5 mg of total protein and rotated overnight at 4°C. 50 μl Protein A Sepharose bead slurry was then added and rotated for two hours at 4°C. To immunoprecipitate FLAG-tagged traptamers, 50 μl of anti-FLAG M2 matrix gel (Sigma-Aldrich, Ronkonkoma, NY) was added to 0.5 mg of total protein and rotated overnight at 4°C.

Immunoprecipitated samples were washed four times with 1 mL NET-N buffer (100 mM NaCl, 0.1 mM EDTA, 20 mM Tris-HCl pH 8.0, 0.1% Nonidet P-40) supplemented with protease and phosphatase inhibitors as above, pelleted and re-suspended in 2x Laemmli sample buffer (2x SB) with 200 mM dithiothreitol (DTT) and 5% β-mercaptoethanol (β-ME). Precipitated proteins and whole cell lysates were heated at 95°C for 5 min and then resolved by SDS-PAGE on either 10%, 15% or 20% polyacrylamide gels according to the size of the protein. The resolving gel was then transferred by electrophoresis to a 0.2 μm nitrocellulose membrane. SDS was added to the transfer buffer in order to detect phosphorylated proteins.

Membranes were blocked with gentle agitation for one hour at room temperature in 5% nonfat dry milk/TBST (1X Tris buffered saline plus 0.1% Tween-20). To detect phosphorylated EPOR, a mouse anti-phosphotyrosine monoclonal antibody PY100 (Cell Signaling) was used at a 1:1000 dilution; to detect phosphorylated JAK2, a rabbit anti-phospho-JAK2 monoclonal antibody (Tyr1008) (clone D4A8, Cell Signaling) was used at 1:1000 dilution; to detect total JAK2, a rabbit anti-JAK2 monoclonal antibody (clone D2E12, Cell Signaling) was used at 1:1000 dilution; to detect HA-tagged EPORs, an HRP-conjugated mouse anti-HA (clone 6E2, Cell Signaling) was used at 1:1000 dilution. Membranes were incubated overnight with gentle agitation in primary antibody at 4°C, washed five times in TBST, and then incubated with gentle agitation for one hour at room temperature in donkey anti-mouse or donkey anti-rabbit HRP (Jackson Immunoresearch, West Grove, PA), as appropriate, at a 1:10,000 dilution. To reprobe phospho-JAK2 and PY100 blots, membranes were stripped in Restore Western Stripping Buffer (ThermoFisher Scientific) for 8 min at room temperature with gentle agitation, washed five times in TBST, blocked in 5% milk/TBST for one hour at room temperature, and incubated overnight at 4°C with anti-JAK2 or anti-HA-HRP antibody, respectively, as described above. Membranes were incubated with Super Signal West Pico or Femto Chemiluminescent Substrates (Pierce ThermoFisher) to detect protein bands.

To assess EPOR oligomerization, we first used immunoblotting with anti-HA to document that full-length-hEPOR and Δ259 were expressed in the appropriate cell lines (*Figure 3D*, bottom panel). To confirm the specificity of the C-20 antibody, detergent lysates of BaF3/hEPOR, BaF3/Δ259 and BaF3/Δ259-hEPOR cells, all lacking traptamer expression, were immunoprecipitated by C-20 antibody, subjected to gel electrophoresis and immunoblotted with anti-HA. Full-length hEPOR was immunoprecipitated whether or not the cells co-expressed Δ259 (lanes 1 and 3 of *Figure 3D*, top panel), but a minimal amount of Δ259 was precipitated by C-20, even when hEPOR was co-expressed (lanes 2 and 3 of *Figure 3D*, top panel).

## Expression and purification of peptides for NMR studies

DNA sequences encoding the TMD segment of human EPOR (residues Leu217 to Gly277 [hEPOR$_{217-277}$], of which residues 226 to 247 constitute the TMD) or human GHR (residues 238 to 274, of which the residues 247 to 270 constitute the TMD) were synthesized and cloned into the pGEX-4T1 vector carrying an N-terminal GST protein and a thrombin cleavage site. FLAG-tagged LH5 traptamer and its mutants LH5-L9I and LH5-I10L were cloned into the same vector with an additional TEV cleavage site in the order: GST-thrombin cleavage site-TEV cleavage site- FLAG-tag-transmembrane sequence. Plasmids were transformed into *E. coli*, and the cells grown in either unlabeled LB medium (for traptamer constructs) or in $^{15}$N-labelled M9 medium (for receptor TMD constructs) supplemented with 100 μg ampicillin to a cell density of 0.5 at OD$_{600}$. Protein expression was induced with 1 mM isopropyl β-D-1-thiogalactopyranoside (IPTG), and cells were harvested 4 hr after induction. All four fusion-proteins were expressed into inclusion bodies, which were harvested by

three cycles of sonication and centrifugation. The inclusion bodies were washed in 50 mM Tris-HCl, pH 7.4 and solubilized in 1.5% (w/v) N-lauroylsarcosine and 100 mM DTT in 50 mM Tris-HCl, pH 7.4. After gentle agitation overnight the insoluble material was removed by centrifugation at 12000 x g and the solubilized proteins dialyzed two times against four L of 0.5% (w/v) N-lauroylsarcosine in 50 mM Tris-HCl, pH 7.4. GST was cleaved off with thrombin (3 units/µL) and the released peptides purified utilizing a chloroform/methanol extraction as described (Bugge et al., 2015), dried under a continuous flow of $N_2$ gas and stored at $-20°C$ until use.

## NMR spectroscopy and titrations

All NMR spectra were recorded at 37°C on a 750 MHz AVANCE Bruker spectrometer equipped with a cold probe. Spectra for backbone assignments of hEPOR$_{217-277}$ (HNCO, HNCAHC, HNCA, HNCACB, CBCACONH, HSQC) were recorded on 1 mM hEPOR$_{217-277}$ solubilized in 40 mM dihexanoylphosphatidylcholine (DHPC), 10% (v/v) $D_2O$, 2 mM tris (2-carboxyethyl)phosphine (TCEP), 1 mM 4,4-dimethyl-4-silapentane-1-sulfonic acid (DSS), 0.05% (v/v) $NaN_3$, 50 mM NaCl, and 20 mM $Na_2HPO_4/NaH_2PO_4$, pH 7.4. Proton chemical shifts were referenced internally to DSS at 0.00 ppm, heteronuclei by their relative gyromagnetic ratios. Monomer-dimer equilibrium-induced chemical shift changes of hEPOR$_{217-277}$ at different DHPC concentrations were derived from comparing HSQC spectra of 0.5 mM hEPOR$_{217-277}$ in 25 mM DHPC giving a ratio of 1:30 of EPOR:DHPC to spectra of 0.5 mM hEPOR$_{217-277}$ in 40 mM DHPC giving a ratio of 1:60 of EPOR:DHPC. To achieve different ratios of traptamer to hEPOR$_{217-277}$ and DHPC, sample conditions were adjusted to 0.5 mM hEPOR$_{217-277}$ in 25 mM DHPC for (0:2):30 [molar ratios of (traptamer:hEPOR):DHPC], 0.25 mM traptamer and 0.5 mM $^{15}N$ hEPOR$_{217-277}$ in 32.5 mM DHPC at (1:2):30 ratios, and 0.5 mM traptamer and 0.5 mM $^{15}N$ hEPOR$_{217-277}$ in 40 mM DHPC at (2:2):30 ratios. HSQC NMR spectra of hGHR$_{238-274}$ without and with LH5-L9I (ratio 4:2) were recorded on 0.25 mM $^{15}N$ hGHR$_{238-274}$ in 25 mM DHPC and on 0.25 mM $^{15}N$ hGHR$_{238-274}$ and 0.5 mM traptamer in 55 mM DHPC, respectively. All NMR samples contained 10% (v/v) $D_2O$, 2 mM TCEP, 1 mM DSS, 0.05% (v/v) $NaN_3$, 50 mM NaCl, and 20 mM $Na_2HPO_4/NaH_2PO_4$, pH 7.4. Chemical shift changes were extracted from $^1H,^{15}N$-HSQC spectra and calculated considering $^{15}N$ and $^1H$ shift dispersions using the following formula: $\delta_{NH} = \sqrt{(\delta_H)^2 + 0.1(\delta_N)^2}$

In all replicate NMR experiments on different batches of peptides, the chemical shift differences induced by the traptamers did not vary more than a maximum of 0.04 ppm between data sets.

To achieve high ratios of LH5 to hEPOR$_{217-277}$ in the titrations, sample conditions were adjusted to 0.33 mM hEPOR$_{217-277}$ in 30 mM DHPC for (0:1):60 [molar ratios of (LH5:hEPOR):DHPC], 0.33 mM LH5 and 0.33 mM $^{15}N$ hEPOR$_{217-277}$ in 50 mM DHPC at (1:1):60 ratios, 0.66 mM LH5 and 0.33 mM $^{15}N$ hEPOR$_{217-277}$ in 70 mM DHPC at (2:1):60 ratios, and 1.32 mM LH5 and 0.33 mM $^{15}N$ hEPOR$_{217-277}$ in 110 mM DHPC at (4:1):60 ratios. Finally, $^1H,^{13}C$-HSQC spectra of $^{13}C,^{15}N$-hEPOR$_{217-277}$ were recorded on 0.5 mM $^{13}C,^{15}N$ hEPOR$_{217-277}$ prepared in 25 mM DHPC to achieve molar ratios of (LH5:hEPOR):DHPC of (0:1):30, and 0.5 mM LH5 and 0.5 mM $^{13}C,^{15}N$ hEPOR$_{217-277}$ in 40 mM DHPC for ratios of (1:1):30.

## Structural modeling

Molecular dynamics simulations were run in CHARMM 38b2 (Brooks et al., 2009) using the IMM1 implicit membrane solvation model (Masunov and Lazaridis, 2003). Simulations of EPOR TMD dimers (hEPOR: LILTLSLILVVILVLLTVALL; mEPOR: LILTLSLILVLISLLLTVLALL) were initialized as ideal coiled-coils in the **VI** register, inserted normal to the membrane. Ten 100-ns simulations, with different random seeds, were then run from this starting conformation, with trajectory snapshots saved every 10 ps. To maximize productive sampling of dimeric states, a gentle harmonic restraint was applied to the distance between helical centroids with a force constant of 0.01 kcal/mol/Å$^2$ and an equilibrium distance of 10 Å. No symmetry restraints were applied in simulations. To prevent artifacts at helical termini associated with the lack of the full-length protein context, backbone i-i +4 hydrogen bonding geometries were restrained for one residue at either terminus of each helix. The restraint was applied to the distance between corresponding backbone N and O atoms with a force constant of 1 kcal/mol/Å$^2$ and the equilibrium distance of 2.87 Å$^2$. Analysis of simulation trajectories revealed rapid sampling of helical phases with frequent switches between states (in fact, the initial **VI** state was rarely visited). The first 10 ns of each trajectory were treated as equilibration and discarded

in all subsequent analysis. Starting conformations for simulating EPOR TMD dimers in the presence of traptamers were generated by extracting random EPOR TMD snapshots from the above trajectories and adding one, two, or four traptamer chains as ideal helices inserted anti-parallel to the EPOR TMD dimer. Traptamer chains were added with random displacements in the plane of the membrane within a radius of 20 Å from the EPOR TMD centroid. For each traptamer/stoichiometry/EPOR type, 100 such random starting points were generated and each used to initialize a 10-ns simulation with trajectory snapshots saved every 10 ps. To ensure proper equilibration of the larger traptamer/EPOR systems, the first 5 ns of each of these simulations were discarded in all analyses. The same restraints at helical termini as described above were applied to all traptamer chains. In addition, to prevent traptamers from diffusing away (and resulting in unproductive sampling), a flat-bottom semi-harmonic potential was applied to keep each traptamer within 20 Å of an EPOR TM helix. This constraint did not contribute energy if the traptamer centroid was within 20 Å of an EPOR helix centroid, while outside this radius the restraint energy increased harmonically (applied to the distance by which the 20 Å limit was exceeded) with a force constant of 1 kcal/mol/Å$^2$. In the 2:2 and 4:2 topologies, half of the traptamer chains were restrained with respect to one EPOR helix and the other half with respect to the other helix.

To produce helical phase distributions, EPOR TM dimer structures from 10,000 randomly chosen trajectory snapshots for each simulation type were subjected to coiled-coil parameter analysis using the CCCP program (*Grigoryan and Degrado, 2011*). CCCP finds an ideal coiled-coil structure that best fits to any given helical bundle. Snapshots that fit with root mean square deviation (RMSD) >0.8 Å were ignored as not representing conformations sufficiently close to a coiled-coil to warrant clear interpretation of resulting parameters (~7,000–8,000 snapshots per simulation type survived this filter). Parameter values from remaining snapshots were used to construct the plots in *Figure 7A*.

To study the rate at which the EPOR dimer may symmetrize, we analyzed all 100 independent implicit-solvent trajectories run for each EPOR/traptamer combination. To limit the effect of the starting conformation, the first 5 ns was discarded as equilibration, with the remainder used to study the extent to which the EPOR dimer would be expected to average out to a symmetric structure over time (i.e., an aggregate of 500 ns were analyzed). Because averaging the structure itself does not yield geometrically realistic conformations, we instead used a distance matrix-based measure of symmetry. We computed the distance matrix $M_i$ corresponding to the EPOR part of each simulation snapshot $i$ (defined as the matrix of distances between all pairs of atoms), and further computed running averages of this matrix $M_n$ over progressively longer time windows of $n$ snapshots, starting from the initial structure (i.e., the mid-point of the original simulation). Next, the degree of symmetry of $M_n$ was calculated as $\sigma_n = \sum_{i<j}^{N} \left| \frac{1}{M_n(i,j)} - \frac{1}{M_n(i',j')} \right|$, where $N$ is the total number of atoms in the EPOR dimer, and $i'$ is the pseudo-symmetry equivalent counterpart of atom $i$ (i.e., if $i$ belongs to the first chain of the EPOR dimer, then $i'$ is the equivalent atom on the second chain; and vice versa).

## All-atom molecular dynamics simulations

The initial atomic co-ordinates for the complexes of the mEPOR and hEPOR TMDs and LH5 were obtained from representative 2:2 centroid models from the implicit-membrane simulations. Complexes with LH5-L9I and LH5-I10L mutant traptamers were generated by starting with the corresponding LH5 complex and performing the necessary amino-acid substitution using the Rosetta Design modelling platform (*Huang et al., 2011*). These models were used as a starting point for all-atom simulations after adding short juxtamembrane domains to each end of the receptor TMDs. The final sequences of the EPOR segments used for modeling were as follows, with the TMDs underlined:

hEPOR MSEPASLLTASDLDP<u>LILTLSLILVVILVLLTVALL</u>SHRRTLQQKWIP
mEPOR MSEPASLLTASDLDP<u>LILTLSLILVLISLLLTVALL</u>SHRRTLQQKWIP

These receptor-traptamer complexes were embedded in 1-palmitoyl-2-oleoyl-sn-glycero-3-phosphocholine (POPC) bilayers containing 128 lipids per bilayer leaflet using CHARMM-GUI (*Jo et al., 2008*). The system was inserted in the center of the membrane perpendicular to the lipid bilayer plane. The system was solvated using TIP3P water molecules, with Na+ and Cl− ions included to maintain neutral charge at physiological salt concentration of 0.154 M.

The all-atom MD simulations were carried out with the Gromacs 2016.1 platform using in-house computing and the Grace high performance computers at Yale University. The AMBER99SB-ILDN-

GORD_SLipids force field (*Lindorff-Larsen et al., 2010*) force-field was used after adding the parameters for POPC molecules as established by SLIPIDS (*Jämbeck and Lyubartsev, 2012*). Energy minimization was carried out in Gromacs with double precision to permit the system to stabilize. Following the minimization, an NVT equilibration was undertaken by applying a restraint on the lipids to allow the water molecules to re-orient around the lipid headgroups and exposed parts of the protein. Next, NPT equilibration was then carried out in two steps using the Nosé-Hoover thermostat (*Nosé, 1984*) and Parinello-Rahman (*Parrinello and Rahman, 1981*) semi-isotropic coupling with a time constant of 2 fs and a temperature of 293°K. Long-range interactions were treated with the particle-mesh Ewald method (*Darden et al., 1993*) (interpolation order of 4 with a grid spacing of 0.12 nm). Initially the protein was kept under restraint and the bilayer was allowed to equilibrate around the protein for 5 ns, following which a restraint on the bilayer was applied and the protein was allowed to equilibrate for 2 ns. The pressure progression and lateral area of the membrane was checked after the NPT equilibration to ensure system stability. The above protocol was followed for each of the prepared system, and these were set for a production MD run of 50 ns without any restraint on the water molecules, ions, protein or the bilayer. The analysis of the MD trajectories was also carried out with Gromacs 2016.1 and the data were plotted using MATLAB (*The MathWorks I, 2000*).

## Acknowledgements

We thank Lisa Petti, Anne Edwards, Anne Bendsoe, and Katrine Bugge for helpful discussions, and Jan Zulkeski for assistance in preparing this manuscript. This work was supported by a grant from the NIH to DD (CA037157), from the Lundbeck and Novo Nordisk Foundations to BBK, and from the NIH and NSF to GG (P20 GM113132 and MCB151032, respectively).

## Additional information

### Funding

| Funder | Grant reference number | Author |
| --- | --- | --- |
| National Institutes of Health | R01 CA037157 | Daniel DiMaio |
| Lundbeckfonden | | Birthe B Kragelund |
| Novo Nordisk Foundation | | Birthe B Kragelund |
| National Institutes of Health | GM113132 | Gevorg Grigoryan |
| National Science Foundation | MCB151032 | Gevorg Grigoryan |

The funders had no role in study design, data collection and interpretation, or the decision to submit the work for publication.

### Author contributions

Li He, Helena Steinocher, Conceptualization, Formal analysis, Investigation, Validation, Methodology, Writing—original draft, Writing—review and editing; Ashish Shelar, Conceptualization, Formal analysis, Investigation, Methodology, Writing—review and editing; Emily B Cohen, Resources, Validation, Writing—original draft, Writing—review and editing; Erin N Heim, Resources, Writing—original draft, Writing—review and editing; Birthe B Kragelund, Conceptualization, Supervision, Funding acquisition, Validation, Writing—original draft, Writing—review and editing; Gevorg Grigoryan, Conceptualization, Formal analysis, Funding acquisition, Validation, Methodology, Writing—original draft, Writing—review and editing; Daniel DiMaio, Conceptualization, Supervision, Funding acquisition, Validation, Writing—original draft, Project administration, Writing—review and editing

### Author ORCIDs

Birthe B Kragelund http://orcid.org/0000-0002-7454-1761
Daniel DiMaio http://orcid.org/0000-0002-2060-5977

### Decision letter and Author response

Decision letter https://doi.org/10.7554/eLife.27701.020

Author response https://doi.org/10.7554/eLife.27701.021

## Additional files

### Supplementary files

• Transparent reporting form
DOI: https://doi.org/10.7554/eLife.27701.019

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
