## [Decision Letter]

Thank you for submitting your article "Single Methyl Groups Can Act as Toggle Switches to Specify Transmembrane Protein-protein Interactions" for consideration by *eLife*. Your article has been favorably evaluated by Jonathan Cooper (Senior Editor) and three reviewers, one of whom, Nir Ben-Tal (Reviewer #1), is a member of our Board of Reviewing Editors. The following individual involved in review of your submission has agreed to reveal their identity: Dieter Langosch (Reviewer #3).

The reviewers have discussed the reviews with one another and the Reviewing Editor has drafted this decision to help you prepare a revised submission.

Summary:

In this work, the authors built on their previous approaches to identify protein-based ligands suited to activate cytokine receptors. Focusing on the erythropoetin receptor (EPOR), they applied a previously developed screening protocol whereby a large combinatorial library of transmembrane domains (26 residues long with either Ile or Leu in each position) was selected for candidates that induce ligand-independent cell survival. Indeed, they identified a number of sequences, referred to as traptamers, and went on to functionally characterize them. They demonstrate specificity in terms of human vs murine EPOR, the type of cytokine receptor, and transmembrane domain sequence. In addition, they show that i) the species-dependence of target receptor rests on the sequence of its own transmembrane segment, and ii) that compensatory mutations can be identified rescuing the activation of otherwise non-cognate pairs. Further, they can relate the activating potential of their novel traptamers to the dimerization of the target receptor, and applied NMR spectroscopy to support this claim. Finally, they present molecular models obtained by simulation that suggest a switch of receptor transmembrane helix-helix interface upon association with the activating traptamers.

Opinion: This is a beautiful study where different technical approaches are connected to provide a coherent view of how these novel artificial transmembrane domain ligands can activate those receptors. A main point is that there appears to be exquisite sequence-specificity behind those interactions. Although many transmembrane helix-helix interactions have been mapped now by mutation, few of them appear to be as specific as the ones presented here (this is almost too good to be true!). However, several issues with the NMR studies and simulations, including a seemingly conflict between the molecular model and NMR spectra, should be resolved before the manuscript can be accepted for publication.

Essential revisions:

1) Focusing on the important role of methyl side chain groups in the molecular mechanism of the traptamer activity in elucidation of their biochemical data, the authors base their claims and explanations on quite limited analysis of NMR data (subtle variations of chemical shifts of amide groups in the 15N-HSQC NMR spectra), and mostly on molecular modeling in implicit membrane (which has been performed for EPOR on multiple occasions for the last 15 years). Such a narrow approach does not allow to confidently determine the structure of the complex, or even its stoichiometry. For example, the structural models of the TMD-traptamer complexes presented in Figure 7 are asymmetric, which would be expected to result in a signal doubling in the NMR spectra but not observed in Figure 4.

2) Furthermore, subtle changes of amide group signals reflect mostly the small perturbations of the backbone structure, e.g. helix bending due to the association. The availability of the recombinant 15N/13C-labeled TMD of EPOR allows direct monitoring of perturbations of the individual 13CH3 signals of the certain residues, e.g., using 13C-NMR techniques. A titration set of the high-resolution constant-time 13C-HSQC experiments can be used to demonstrate the 13CH3 signal doubling (even without peak assignments) upon complex formation of TMD EPOR with certain biologically active traptomer.

3) In addition, NMR relaxation experiments can be used for estimation of the complex size. Such experimental structural information accompanied with molecular modeling (preferably in explicit lipid bilayer) could help produce a credible model of the TMD-traptamer complex and explain molecular mechanisms of functioning of not only the traptamers themselves, but possibly some other classes of TM peptides that modulate single-span receptor activation.

4) In principle, the simulations should give a very clear explanation of the effect of moving a methyl group on dimerization. The description in Figure 7 and the relevant section in the main text do not provide such explanation. Which interactions are formed/deformed/deleted upon moving the methyl group?

---

## [Author Response]

Essential revisions:1) Focusing on the important role of methyl side chain groups in the molecular mechanism of the traptamer activity in elucidation of their biochemical data, the authors base their claims and explanations on quite limited analysis of NMR data (subtle variations of chemical shifts of amide groups in the 15N-HSQC NMR spectra), and mostly on molecular modeling in implicit membrane (which has been performed for EPOR on multiple occasions for the last 15 years). Such a narrow approach does not allow to confidently determine the structure of the complex, or even its stoichiometry. For example, the structural models of the TMD-traptamer complexes presented in Figure 7 are asymmetric, which would be expected to result in a signal doubling in the NMR spectra but not observed in Figure 4.

We agree that the NMR data are insufficient to determine the structure or the stoichiometry of the complex, and did not mean to imply that they did. Indeed, our data suggest that there is not a single defined structure of the complex, and rather that it is a dynamic structure in which the traptamers basically compete with the lipid to induce a larger fraction of EPOR to reside in an active conformation. Of course, this makes the specificity the traptamers display and the modulation of their activity by moving a single methyl group all the more interesting. These points are now discussed in the fourth paragraph of the Discussion.

We do not believe that the modeling and the NMR are in conflict, and rather that the apparent discrepancies reflect the different timescales of the experiments. In simulations, homo-dimers are always asymmetric at any given snapshot of time in solution at room temperature, which is what we observed in the modeling. In contrast, if the dimer is symmetric at the timescale of the NMR experiment (i.e., if the structure has time to equilibrate/average over the non-symmetric micro-states to arrive at a symmetric average), then there will be no peak doubling, as was indeed the case for the NMR data. To further address this, we extracted the timescale at which the dimer approaches symmetry in the implicit MD simulations. As shown in new Figure 7—figure supplement 2, the simulations approach symmetry much faster than NMR timescales, providing an explanation for the apparent discrepancy pointed out by the reviewer.

We also performed an additional NMR experiment, titrating increasing amounts of traptamer into the hEPOR TMD and measuring the chemical shifts in response to traptamer addition. As shown in the new Figure 4, we see clear saturation of the NMR signals for residues in the hEPOR TMD. These results provide further evidence that the traptamers bind specifically to the hEPOR TMD and suggest that there are multiple traptamers in association with each hEPOR TMD dimer.

2) Furthermore, subtle changes of amide group signals reflect mostly the small perturbations of the backbone structure, e.g. helix bending due to the association. The availability of the recombinant 15N/13C-labeled TMD of EPOR allows direct monitoring of perturbations of the individual 13CH3 signals of the certain residues, e.g., using 13C-NMR techniques. A titration set of the high-resolution constant-time 13C-HSQC experiments can be used to demonstrate the 13CH3 signal doubling (even without peak assignments) upon complex formation of TMD EPOR with certain biologically active traptomer.

These very simple traptamers, consisting solely of leucine and isoleucine, pose some unique challenges for NMR, such as the fact that their hydrocarbon amino acid side chains chemically closely resemble the fatty acid tails of the lipid. Similarly, the detergents that are present tend to obscure the spectra to an extent that affects peak resolution. Nevertheless, we have performed 13C NMR as suggested and we see no doubling of signal in accordance with the symmetry of the complex. The addition of the traptamers did induce the appearance of a few new as-yet-unassigned EPOR side chain peaks supporting the conclusion that the traptamers and the EPOR TM region interact directly. This is shown in new Figure 4—figure supplement 1.

3) In addition, NMR relaxation experiments can be used for estimation of the complex size. Such experimental structural information accompanied with molecular modeling (preferably in explicit lipid bilayer) could help produce a credible model of the TMD-traptamer complex and explain molecular mechanisms of functioning of not only the traptamers themselves, but possibly some other classes of TM peptides that modulate single-span receptor activation.

Measuring NMR relaxation to estimate complex size is complicated because the addition of detergent needed to keep the protein:detergent ratio fixed critically changes the overall sample viscosity. This impacts tumbling rate, which makes it impossible to distinguish between a viscosity effect and a size effect, as discussed in the literature (Stanczak et al., JACS, 131(51): 18450-59, 2009). Furthermore, the presence of the micelles (DHPC micelle size of 4-5 nm), together with the very small size of the peptides, makes size determination unreliable. Thus, we do not believe it is possible to make a credible size determination in this system.

As suggested by the reviewer, we have performed all-atom MD simulations of the traptamer/EPOR TMD complex in an explicit lipid bilayer, with an aggregate of 100 ns of production sampling per system. Although this used a considerable amount of computational resources, these simulations are not long enough to produce a concrete model of structural modulation by traptamers. Indeed, developing an accurate model of action for a novel transmembrane protein purely from MD would be essentially unprecedented. Compounding the challenge here is the lack of clear evidence as to the specific traptamer/EPOR TMD stoichiometry. Nevertheless, we used explicit-solvent modeling to confirm and extend several of our conclusions from the implicit-solvent simulations. For example, we show that the complexes remain intact in the presence of lipid even in the absence of association restraints and that the traptamers can have dramatic effects on the structure and stability of the complex. These results are now shown in a new figure, Figure 8.

4) In principle, the simulations should give a very clear explanation of the effect of moving a methyl group on dimerization. The description in Figure 7 and the relevant section in the main text do not provide such explanation. Which interactions are formed/deformed/deleted upon moving the methyl group?

As noted above, we have begun all-atom simulations to address this question. We are not yet able to assign specific consequences of moving the methyl group, but we have confirmed that moving these groups has a dramatic effect on the RMSD and RMSF of the hEPOR complexes, suggesting changes in dynamics.

Interestingly, these changes spread out along the entire TMD. These results are also shown in Figure 8. In addition, in the implicit membrane modeling, we have further analyzed the movements of the side chains of the relevant amino acids (position 9 and 10 in the traptamer and position 238 of the EPOR), and now display the data as a rosette plot in Figure 7. We think that this is a clearer representation of the complex, because it highlights its dynamic character, and we have replaced the static structure in the original Figure 7 with it.